# Neurovascularization inhibiting dual responsive hydrogel for alleviating the progression of osteoarthritis

Wenpin Qin [1,2,3,4,8], Zhangyu Ma[1,2,3,4,8], Guo Bai[5,6,7,8], Wen Qin[2,3,4], Ling Li[2,3,4], Dongxiao Hao[2,3,4], Yuzhu Wang[2,3,4], Jianfei Yan[2,3,4], Xiaoxiao Han[1], Wen Niu[2,3,4] ✉, Lina Niu [2,3,4] ✉ & Kai Jiao [1,2,3,4] ✉

Treating osteoarthritis (OA) associated pain is a challenge with the potential to significantly improve patients lives. Here, we report on a hydrogel for extracellular RNA scavenging and releasing bevacizumab to block neurovascularization at the osteochondral interface, thereby mitigating OA pain and disease progression. The hydrogel is formed by cross-linking aldehyde-phenylboronic acid-modified sodium alginate/polyethyleneimine-grafted protocatechuic acid (OSAP/PPCA) and bevacizumab sustained-release nanoparticles (BGN@Be), termed OSPPB. The dynamic Schiff base bonds and boronic ester bonds allow for injectability, self-healing, and pH/reactive oxygen species dual responsiveness. The OSPPB hydrogel can significantly inhibit angiogenesis and neurogenesis in vitro. In an in vivo OA model, intraarticular injection of OSPPB accelerates the healing process of condyles and alleviates chronic pain by inhibiting neurovascularization at the osteochondral interface. The injectable hydrogel represents a promising technique to treat OA and OA associated pain.

Osteoarthritis (OA) is the most common musculoskeletal disorder, with significantly high morbidity[1]. It affects more than 595 million people worldwide, leading to a significant health-system burden[1]. Osteoarthritis commonly causes pain, disability, and loss of function. To treat OA, pharmacotherapy, physiotherapy, and exercise are widely used. Pharmacotherapy often serves as an initial therapeutic intervention, aiming to promptly mitigate acute symptoms, while the latter two modalities play a complementary and sustaining role in the overall treatment strategy for OA. Pharmacotherapy includes non-steroidal anti-inflammatory drugs, paracetamol, opioids, and corticosteroids, focusing on the inhibition of inflammation and alleviating pain, but not addressing the pathological changes. Inevitably, the ultimate outcome is joint replacement surgery[1]. Currently, the most compelling evidence for pain and pathological changes centers on neurovascularization at the osteochondral interface[2]; however, therapeutic and analgesic strategies specifically targeting neurovascularization have been notably underutilized[3]. A large number of nerves and vessels bud from the subchondral bone, penetrate the osteochondral interface, and invade avascular cartilage through vertical microcracks[4]. This penetration transports inflammatory mediators to the cartilage and accelerates

[1]Department of Stomatology, Tangdu Hospital, The Fourth Military Medical University, Xi'an 710000 Shaanxi, China. [2]State Key Laboratory of Oral and Maxillofacial Reconstruction and Regeneration, The Fourth Military Medical University, Xi'an 710000 Shaanxi, China. [3]National Clinical Research Center for Oral Diseases, The Fourth Military Medical University, Xi'an 710000 Shaanxi, China. [4]Shaanxi Key Laboratory of Stomatology, School of Stomatology, The Fourth Military Medical University, Xi'an 710000 Shaanxi, China. [5]Department of Oral Surgery, Shanghai Ninth People's Hospital, Shanghai Jiao Tong University School of Medicine, Shanghai 200011, China. [6]College of Stomatology, Shanghai Jiao Tong University, Shanghai 200011, China. [7]National Center for Stomatology, National Clinical Research Center for Oral Diseases, Shanghai Key Laboratory of Stomatology, Shanghai Research Institute of Stomatology, Research Unit of Oral and Maxillofacial Regenerative Medicine, Chinese Academy of Medical Sciences, Shanghai 200011, China. [8]These authors contributed equally: Wenpin Qin, Zhangyu Ma, Guo Bai. ✉e-mail: niuwen9302@fmmu.edu.cn; niulina831013@126.com; kjiao1@163.com

degradation and mineralization of the matrix, leading to condylar osteophyte formation and aggravation of OA-related pain[5]. It also promotes mechanical and chemical osteoclast-chondrocyte crosstalk, which prompts abnormal subchondral bone remodeling and pain[3]. Thus, neurovascularization at the osteochondral interface is a potential target for OA intervention.

Currently, several cytokines, such as tumor necrosis factor α and interleukin-1β, as well as growth factors, including nerve growth factor[6], vascular endothelial growth factor (VEGF)[7], and platelet-derived growth factor BB[8], have been found to promote OA neurovascularization. In addition, previous research revealed that extracellular RNA (exRNA) serves as an organic polyanionic recruiter[9,10] that binds to and potentiates the function of polycationic neurovascular factors in OA[11]. Further research also verified that intra-articular injection of nucleic acids can induce arthritis via pannus formation[12], and that controllable release of RNA can promote angiogenesis and osteogenesis[13,14]. Similarly, the pivotal role of exRNA in neurovascularization is also found in cardiovascular diseases[15,16], which can modify the function of VEGF[17,18]. Based on these facts, an exRNA-scavenging strategy might outperform mere inhibition of neurovascular factors in OA pain management. To date, exRNA scavenging strategies have been widely used to cure diseases[19] such as periodontitis[20], inflammatory bowel disease[21], obesity[22], severe sepsis[23], psoriasis[24], acute kidney injury[25], and rheumatoid arthritis[26–29]. Despite the demonstrated effectiveness of this strategy in various diseases, the challenge of designing and realizing the responsive release of polycationic agents that can bind exRNA to mitigate toxicity remains a subject for further investigation.

Responsive hydrogels, which facilitate environmentally triggered drug release, have been extensively studied in OA treatment, promising to realize the responsive release of polycationic agents and mitigate toxicity[30]. The substantial changes to the osteochondral microenvironment in OA are regarded as an ideal condition to realize controlled drug delivery. Specifically, the reactive oxygen species (ROS) level is significantly increased in the joints of patients with OA, and the pH of the synovial fluid and articular cartilage is as low as 6.0 or 6.3, respectively[31]. When subjected to acidic and oxidative stress, stimulus-responsive bonds, exemplified by Schiff base bonds and boronic ester bonds[32], undergo cleavage, leading to the disintegration of hydrogel networks and the subsequent release of encapsulated drugs. We anticipate that a dual responsive hydrogel will facilitate the release of polycationic agents and inhibit neurovascularization at the osteochondral interface, which remains a significant challenge.

The temporomandibular joint (TMJ) is particularly vulnerable to OA, with an observed incidence rate of 14.56% in individuals under 30 years old, increasing to 28–32% in people aged between 30 and 80[33]. Previous research confirmed that pathological neurovascularization at the osteochondral interface of condyles in TMJOA[11] is closely associated with maxillofacial pain. To inhibit TMJOA and alleviate pathological neurovascularization, we construct the OSPPB hydrogel system, formed by aldehyde-phenylboronic acid-modified sodium alginate/ polyethyleneimine-grafted protocatechuic acid (OSAP/PPCA) with bioactive glass nanoparticles (BGN) loaded with bevacizumab (BGN@Be). The OSPPB hydrogel has the following advantages: (1) The polycationic OSPPB hydrogel could scavenge polyanionic exRNA; (2) its pH/ROS dual responsiveness realizes the on-demand release of active ingredients; and (3) sequential delivery of active ingredients promotes structural restoration (Fig. 1). Subsequently, we evaluate its inhibitory function on neurons and endothelial cells in vitro, and investigate its impact on condylar neurovascularization and continuously aggravated pain in a mouse unilateral-anterior-crossbite TMJOA model[34–36]. This investigation posits that OSPPB hydrogel scavenges exRNA, inhibits neurovascularization, and rescues the progression of TMJOA.

## Results

### Preparation and characterization of OSAP, PPCA, and BGN@Be
The fabrication process of the OSPPB hydrogel is shown in Fig. 2a. To construct the OSPPB hydrogel, we cross-linked the aldehyde-phenylboronic acid-modified sodium alginate (OSAP) to the polyethyleneimine-grafted protocatechuic acid (PPCA) molecular backbone via the formation of dynamic boronic ester bonds and Schiff base bonds, and added BGN@Be through coordination interactions. The boronic ester bonds were established by crosslinking of the OSAP boronic acid groups with the PPCA catechol groups. Concurrently, the PPCA amino groups were crosslinked with the OSAP aldehyde groups to form Schiff base bonds.

3-aminobenzeneboronic acid was grafted onto sodium alginate to yield the SAP polymers (Fig. 2a). Subsequently, sodium periodate was used to oxidize the hydroxyl groups of SAP to obtain the OSAP polymers, with boronic acid groups and aldehyde groups. Attenuated total reflection-Fourier transform infrared spectroscopy (ATR-FTIR) (Supplementary Fig. 1) confirmed the construction of the OSAP polymers. The peaks at 708 cm$^{-1}$, 1485 cm$^{-1}$, 1339 cm$^{-1}$, and 1735 cm$^{-1}$ indicated the C−H bending vibration of m-substituted benzene, phenyl groups, B−O vibration, and the aldehyde group, respectively[37,38]. The PPCA polymer was synthesized via an amidation reaction, which involved the conjugation of carboxyl groups from protocatechuic acid with amino groups present in polyethyleneimine (Fig. 2a). The chemical structures of the PPCA polymers were tested using ATR-FTIR spectroscopy (Supplementary Fig. 2). The band at 1556 cm$^{-1}$ was attributed to N-H bending of amine, and a new absorption peak at 1643 cm$^{-1}$ represented the carboxamide I band. The as-prepared BGN exhibited a uniform and spherical morphology, with a relatively homogeneous particle size (~240 nm) (Supplementary Fig. 3a, b). Subsequently, BGN@Be was fabricated based on the coordination interactions between Ca$^{2+}$ and bevacizumab. The transmission electron microscopy images clearly revealed the beads-on-a-string structure of BGN@Be, indicating that bevacizumab was successfully wrapped onto BGN (Supplementary Fig. 3a). Nanoparticle tracking analysis revealed its bigger size (approximately 424 nm) and X-ray diffraction confirmed that bevacizumab grafting did not affect the structure of BGN (Supplementary Fig. 3b, c). After soaking in phosphate-buffered saline (PBS) for 21 days, the BGN@Be became very loose (Supplementary Fig. 3a), indicating its degradability. The amounts of Si (89% within 10 days, followed by a plateau phase) and bevacizumab (90% within 7 days, followed by a plateau phase) from BGN@Be increased significantly with soaking time, confirming its slow release performance (Supplementary Fig. 3d, e).

### OSPPB hydrogel preparation and characterization
To obtain the OSPPB hydrogel, BGN@Be-dispersed OSAP solution and BGN@Be-dispersed PPCA solution were mixed by vortexing. The hydrogel formed within 3 s, due to the formation of dynamic boronic ester bonds and Schiff base bonds (Fig. 2b). The chemical structure of the hydrogel was confirmed using ATR-FTIR spectroscopy. The new peak at 1431 cm$^{-1}$ of the hydrogel was attributed to the boronic ester bonds[39], and the disappearance of the characteristic peak at 1735 cm$^{-1}$ indicated that the aldehyde groups were involved in Schiff base reactions (Fig. 2c)[38]. Next, the hydrogel microstructure and the distribution pattern of BGN@Be in the hydrogel were assessed using scanning electron microscopy and energy-dispersive spectroscopy. Both the OSPP (composed of OSAP polymers and PPCA polymers) and OSPPB hydrogels exhibited a homogeneous porous three-dimensional structure, with approximately 200 μm micropores (Fig. 2d). However, they differed in their surface morphology, wherein the OSPPB hydrogel was uniquely marked by scattered distribution of BGN on its surface (The red circles in Fig. 2d). In other words, the incorporation of BGN@Be did not influence the porous structure of the hydrogel. Moreover, the presence of elemental Si, Ca, and P in the red circles (areas with

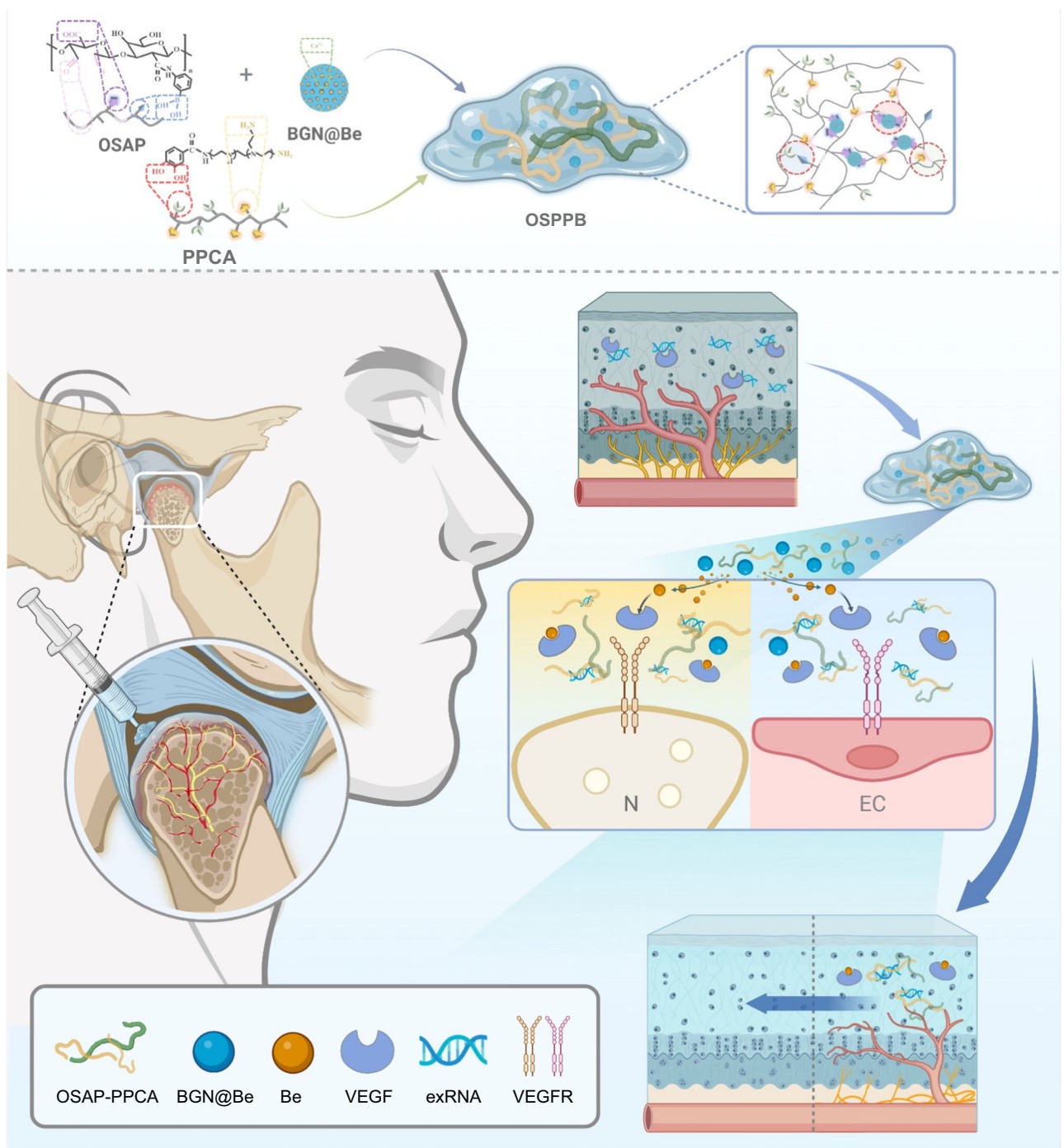

**Fig. 1 | The injectable, self-healing, pH/ROS dual-responsive, polycationic hydrogel reversed TMJOA and alleviated chronic pain by inhibiting neurovascularization at osteochondral interface.** Be bevacizumab, VEGF vascular endothelial growth factor, exRNA extracellular ribonucleic acid, VEGFR VEGF receptor. Created in BioRender.

BGN@Be distribution), besides the absence of them out of red circles of OSPPB hydrogel (areas without BGN@Be distribution) and OSPP hydrogel, further confirmed BGN@Be incorporation into the OSPPB hydrogel (Fig. 2e and Supplementary Fig. 4).

The dynamic Schiff base bonds and boronic ester bonds endowed the OSPPB hydrogel with multiple favorable characteristics, e.g., remolding, injectability, self-healing, and pH/ROS dual-responsiveness. As shown in Fig. 2b 1–2, the OSPPB hydrogel exhibited excellent remodeling properties and injectability. To demonstrate this, the hydrogel was injected using a 26G needle to write the letter "J".

As depicted in Fig. 2b 3–6, we tested its self-healing property. Before the OSPPB hydrogel was cut into two equal parts, it was formed into a cylindrical shape. Then, the tears were reattached for 10 min in 37 °C. The two parts of the hydrogel gradually merged together to form a monolithic hydrogel, which had sufficient mechanical stiffness to be picked up readily using tweezers, without any fractures along the contact interface. Furthermore, we examined the versatile rheological properties of the hydrogels using a rheometer (Fig. 2f, g). Both the OSPP and OSPPB hydrogels showed a steady storage modulus (G') and loss modulus (G") in the frequency sweep sequence and time sweep

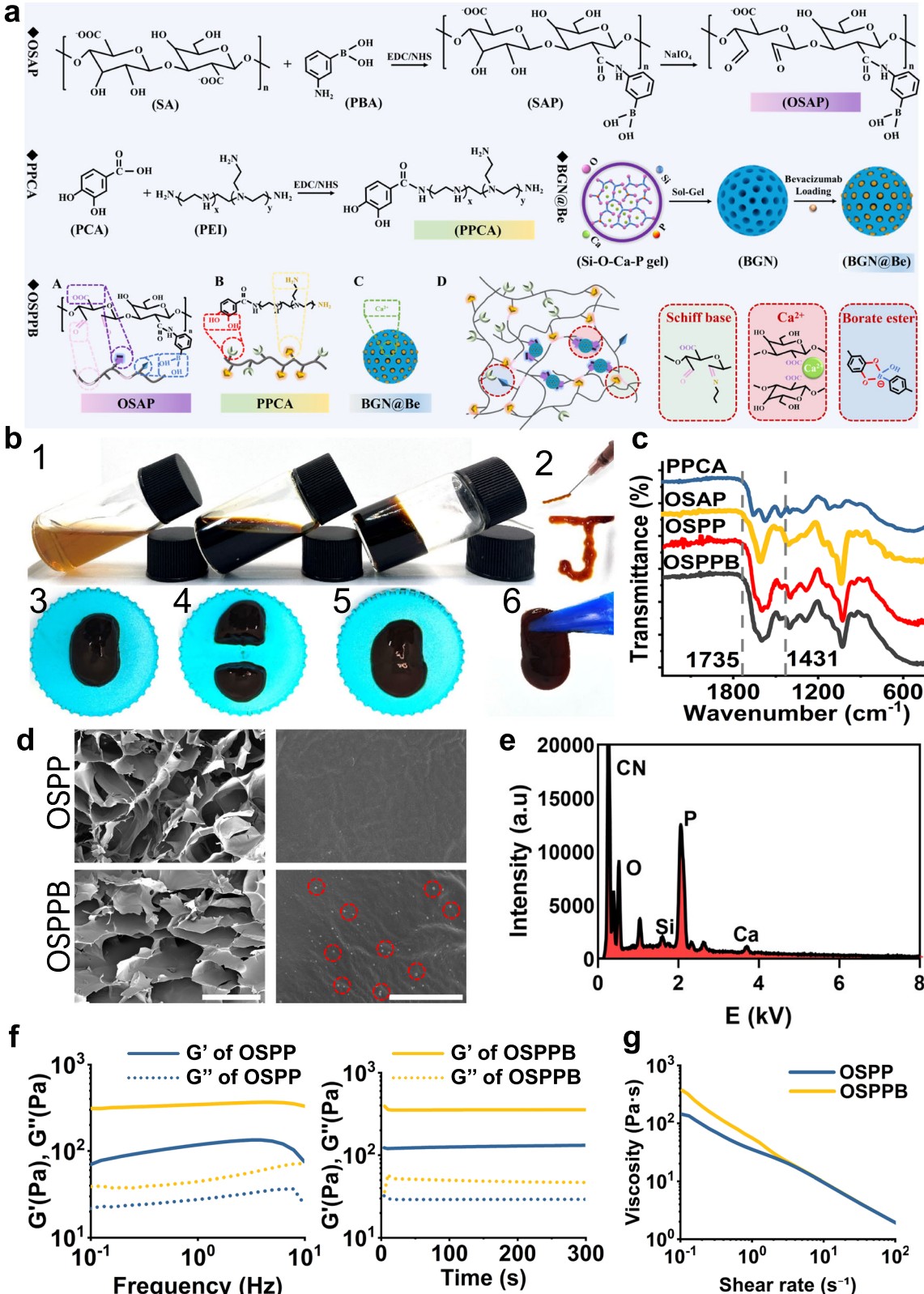

**Fig. 2 | Preparation and characterization of the OSPPB hydrogel. a** The facile fabrication route of the OSPPB hydrogel. **b** The gelation, injectability, and self-healing property of the OSPPB hydrogel. **c** ATR-FTIR of PPCA, OSAP, OSPP, and OSPPB, showing the Schiff base bonds and boronic ester bonds leading to gelation. **d** SEM of OSPP and OSPPB, showing the uniformly distributed BGN@Be on the surface of OSPPB hydrogel. The red circle represents the BGN. Scale bars = 200 μm (left) and 5 μm (right). **e** SEM-EDS of OSPPB hydrogel (areas with BGN distribution), showing the Ca, P and Si elements on OSPPB hydrogel. **f** Frequency-sweep, Time-sweep sequence, and **g** Shear-thinning test of the OSPP and OSPPB hydrogels. G' means storage modulus and G" means loss modulus.

sequence. G′ > G″ confirmed the successful formation of the hydrogel (Fig. 2f). Compared with the lower G′ observed in the OSPP hydrogel (129.46 ± 3.09 Pa), the higher G′ in the OSPPB (354.42 ± 3.97 Pa) confirmed its augmented gel formation characteristics, indicative of a more robust and structurally resilient gel network. The incorporation of BGN@Be into the hydrogel increased its linear viscoelastic zone (Fig. 2f). Besides, the viscosity curves of the two hydrogels were comparable, exhibiting a non-Newtonian shear thinning behavior because the viscosity decreased with increasing shear rate (Fig. 2g). These properties facilitated intra-articular injection, without influencing the hydrogel structure.

Considering its potential pH/ROS dual-responsiveness, we examined the hydrogel pH- and ROS-response variation. Low pH and high ROS environments were simulated using HCl and $H_2O_2$, respectively (Fig. 3a–d). Low pH or high ROS stimulation induced rapid hydrogel collapse and transformation into a liquid-like state (Fig. 3a). The rheological results revealed that there were marked drop in the G′ of the OSPPB hydrogel after low pH (76.76 Pa at 0.1 Hz) or high ROS stimulation (74.53 Pa at 0.1 Hz). And this drop was much more pronounced with both stimuli, with G′ reducing to 17.133 Pa at 0.1 Hz (Fig. 3b). And the SEM results revealed that, while there were still visible pores in the groups subjected to either ROS (31.50% porosity) or pH (37.04% porosity) stimulation alone, the pores in the ROS + pH group (13.08% porosity) had collapsed, indicating network disintegration (Fig. 3c, d). These results provided strong evidence that the OSPPB hydrogel exhibited pH/ROS dual-responsiveness. This pH/ROS dual-responsiveness allowed it to disintegrate completely within 6 days (Fig. 3e). Moreover, the progressive elevation of the sulfur content in the supernatant, exclusively attributed to bevacizumab within the hydrogel, signified a controlled and gradual release of bevacizumab from the hydrogel matrix (Fig. 3f). The dissociation of Schiff base bonds and boronic ester bonds imparted its pH/ROS dual-responsiveness.

To test its hemocompatibility, different concentrations of extracts were incubated with red blood cells. The optical results were similar among the different concentration groups (0 µg/mL to 300 µg/mL), indicating that a safe hemolysis rate (below 5%) was obtained when the concentration was lower than 300 µg/mL (Supplementary Fig. 5). The hydrogel's antioxidant effect was verified using a 2,2-diphenyl-1-picrylhydrazyl (DPPH) scavenging assay. The DPPH scavenging rate of the low-concentration extract was comparable to that of Vitamin C (positive control) (Fig. 3g). Low concentrations were chosen for this experiment because of the inherent dark coloration of both the hydrogel and extracts, thereby minimally impacting the results. However, in the 50 µg/mL group, a marked darkening occurred, which could not be definitively attributed to either an increase in the OSPPB concentration or a decrease in antioxidant capacity. Hence, this experiment only qualitatively confirmed the antioxidant properties of the OSPPB hydrogel. This property might help to reduce the inflammatory response and alleviate oxidative stress in vivo, thus treating OA.

Positive charge and nucleic acid scavenging property are important parameters for this hydrogel. The electrostatic charge determines its nucleic acid scavenging property, since their binding depends on electrostatic interactions[26]. The zeta potential of OSPPB confirmed its positive charge (Fig. 3h). The binding test showed that little Cy3 remained on the dishes after OSPPB treatment, which verified that the OSPPB hydrogel disturbed the binding between VEGF and RNA (Fig. 3i–k). Furthermore, molecular dynamics simulations validated that, within the same system at pH 6.5, PEI exhibited an enhanced propensity for RNA interaction relative to VEGF (Fig. 3l). This notable nucleic acid scavenging property would enable the OSPPB hydrogel to capture the exRNA in osteoarthritic condyles, with the potential to inhibit abnormal neurovascularization.

## OSPPB hydrogel-mediated inhibitory effects on neurovascularization in vitro

Inhibiting the function of peripheral nerves and endothelial cells is the crucial criteria for the OSPPB hydrogel. Hence, the effects of the OSPPB hydrogel on trigeminal ganglion (TG) cells and endothelial progenitor cells (EPCs) were tested in vitro. The LIVE-DEAD staining of EPCs (Supplementary Fig. 6) and the cell counting kit-8 results (Supplementary Fig. 7) showed no variations in cell viability after co-culture with the OSPPB hydrogel. We then evaluated angiogenesis by EPC migration and tube formation, and evaluated neurogenesis by the morphology and axonal growth of TG cells (Fig. 4a, b). Specifically, EPC migration was analyzed by a wound scratch experiment and a Transwell assay. RNA-VEGF treatment shortened the scratch and increased the migrated cell counts. However, treatment with PPCA and BGN@Be displayed a slight inhibitory effect on this process, whereas the OSPPB hydrogel demonstrated a stronger suppressive function (Fig. 4a, c, d). Additionally, treatment with hydrogel caused a blockade of RNA-VEGF-stimulated EPC tube-formation in Matrigel (Fig. 4a). The morphology and axonal growth of TG cells were measured by crystal violet staining and immunofluorescence staining of β3-tubulin. Compared with RNA-VEGF-treated TG cells, cells treated with PPCA, BGN@Be and OSPPB hydrogel exhibited a shorter dendritic length, fewer interactions, and a much less elaborate morphology (Fig. 4b, e, f). Experiments with the hydrogel's diverse components revealed that OSAP lacked inhibitory activity, with PPCA and BGN@Be identified as the key bioactive constituents. Despite PPCA and BGN@Be's inhibitory effects on neurovascularization, their efficacy is surpassed by the better inhibitory capacity of the OSPPB hydrogel. These results suggest that the OSPPB hydrogel could inhibit angiogenesis and neurogenesis in vitro.

## Effect of the OSPPB hydrogel on TMJOA progression in vivo

Next, we constructed a unilateral crossbite (UAC) mouse model[34–36] to evaluate the hydrogel's function on TMJOA in vivo (Fig. 5a). The UAC procedure and intra-articular OSPPB hydrogel injection were conducted in 1 day. Since nonsteroidal anti-inflammatory drugs serve as a first-line treatment for relieving arthritis pain, we used celecoxib administered daily by oral gavage as a positive control. During the whole experiment, all mice survived well without infection. Three weeks after injection, the mice were sacrificed and the condyles and major organs were harvested. Histological analysis showed no inflammatory infiltration or obvious damage in the major organs, including the heart, liver, spleen, lungs, and kidneys (Supplementary Fig. 8). Immunofluorescence showed that the rhodamine B-stained OSPPB hydrogel was distributed to both cartilage and subchondral bone in the UAC group after intra-articular injection (Supplementary Fig. 9). These data proved that the positively-charged OSPPB hydrogel was biocompatible and could overcome the "joint barrier"[31,40] to target the osteochondral interface.

Stereomicroscopic observation identified randomly scattered new vessels in the UAC + Veh and UAC + Celecoxib group, but not in the UAC + OSPPB group (Fig. 5b). The results of hematoxylin and eosin (H&E) and Safranin O/fast green staining further showed the healing effect on TMJ condyles in the OSPPB-treated group, with a lower Osteoarthritis Research Society International (OARSI) score, fewer capillaries, a thicker cartilage layer, and more proteoglycan in the condyles (Fig. 5c1, c2, d–g). Silver staining revealed fewer nerve fibers after OSPPB treatment (Fig. 5c3, h). In addition, the total length of micro-fractures increased in the UAC mice, decreased notably with OSPPB hydrogel treatment, and marginally reduced in the UAC + Celecoxib group (Fig. 5c4, i). Through these fractures, the nerves, vessels, inflammatory factors, and enzymes invade cartilage to establish the "osteoclast–chondrocyte crosstalk", eventually accelerating OA[3]. Three-dimensional reconstruction from micro-computed tomography scans indicated smoother articular surfaces in the OSPPB

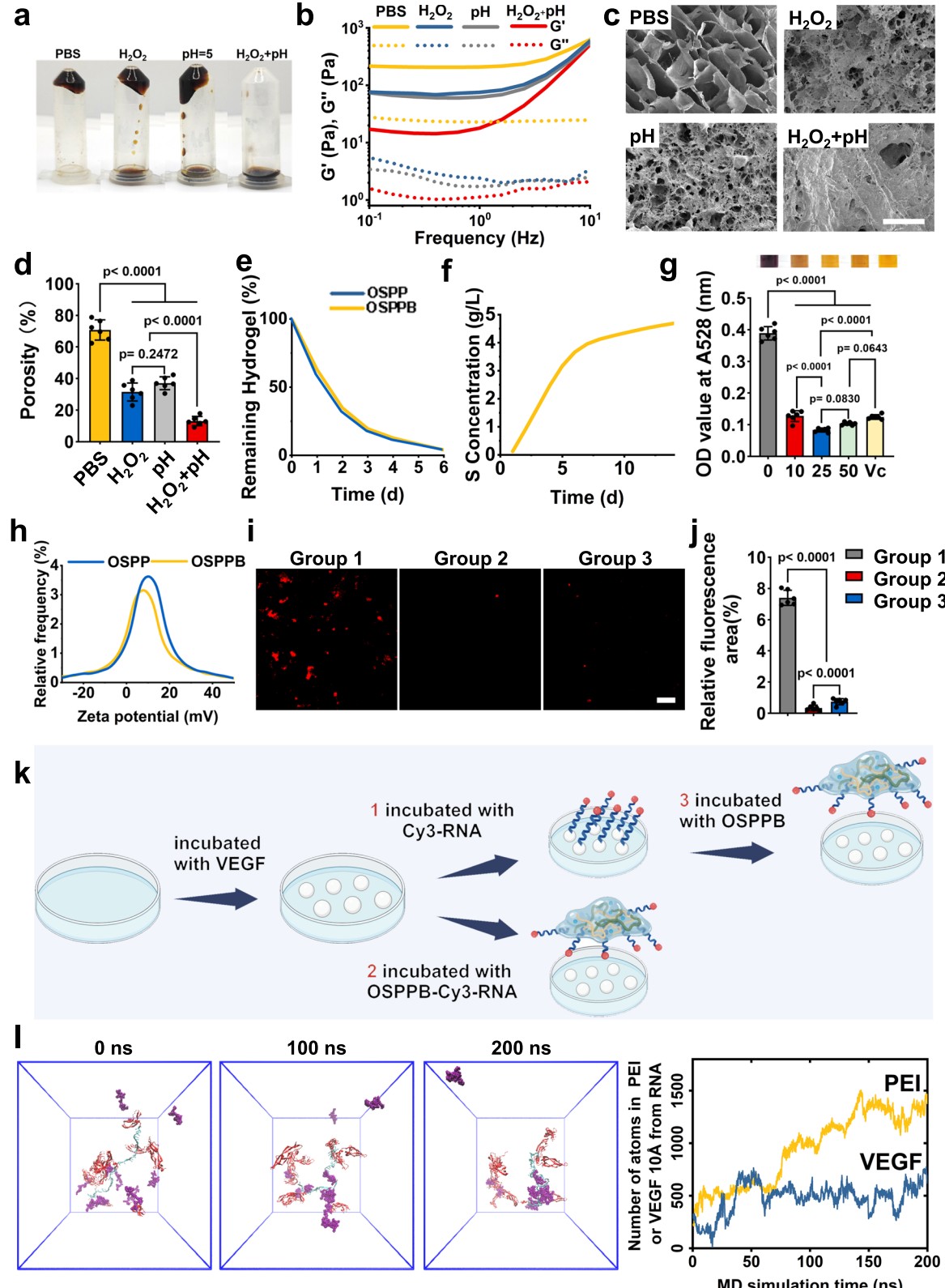

**Fig. 3 | pH/ROS dual responsiveness, degradation, antioxidation property and positive electrical property of the OSPPB hydrogel. a**–**d** Images, frequency-sweep, and SEM (Scale bars = 200 μm) of the OSPPB hydrogel after incubation with PBS (pH = 5) and/or $H_2O_2$ ($n = 6$, $F_{(3, 20)} = 140.7$, $p < 0.0001$). **e** Degradation of OSPP and OSPPB hydrogels in PBS. **f** Characteristics of the S concentration changes over 14 days. **g** Photographs of DPPH after culture with OSPPB for 30 min, with the corresponding statistical results ($n = 6$, $F_{(4, 25)} = 567.7$, $p < 0.0001$). **h** Zeta potential of OSPP and OSPPB hydrogels. **i**–**k** Characteristics of the binding test, showing the binding between OSPPB and RNA ($n = 6$, $F_{(2, 15)} = 930.3$, $p < 0.0001$). Created in BioRender. Scale bar = 50 μm. **l** Molecular dynamic modulation and its quantification for RNA (light blue), VEGF (red) and PEI (Polyethyleneimine, purple). Data are shown as the means and standard deviations. One-way ANOVA is followed by the Tukey-Kramer method for post hoc multiple comparisons.

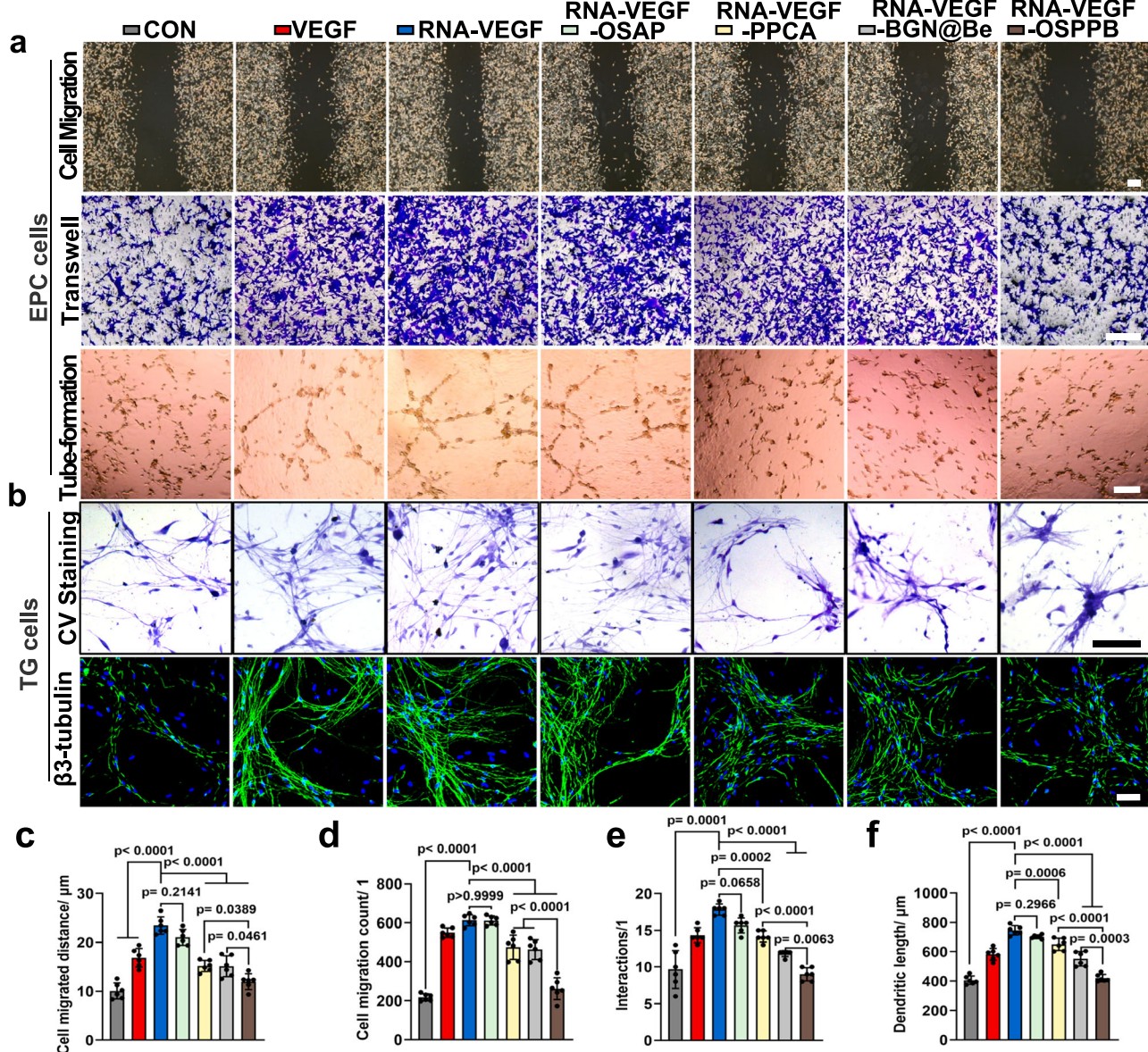

**Fig. 4 | The OSPPB hydrogel inhibits angiogenesis and neurogenesis.**
**a**, **b** Representative microscopy images of EPCs and TG cells showing cell migration, tube formation, and morphology after 24 h of treatment. Scale bars = 100 μm. **c**, **d** Quantification of migrated cells and the migration distance of EPCs after treatment. $n = 6$, $F_{(6, 35)} = 45.9$, $p < 0.0001$ in (**c**). $n = 6$, $F_{(6, 35)} = 89.54$, $p < 0.0001$ in (**d**). **e**, **f** Quantification of the interactions and dendritic length of TG cells after treatment. $n = 6$, $F_{(6, 35)} = 39.66$, $p < 0.0001$ in (**e**). $n = 6$, $F_{(6, 35)} = 85.86$, $p < 0.0001$ in (**f**). Data are shown as the means and standard deviations. One-way ANOVA is followed by the Tukey-Kramer method for post hoc multiple comparisons.

group, without erosion or irregularities (Fig. 5c5). The quantitative results further confirmed the increment of bone mineral density following the application of OSPPB, with Celecoxib showing a less pronounced effect (Fig. 5j, k). Quantitative real-time reverse transcription polymerase chain reaction also proved that injection of the OSPPB hydrogel altered the pro-angiogenic and pro-neurogenic microenvironment in osteoarthritic condyles (Fig. 5l).

The osteochondral interface (approximately 50 μm thick) ensures a smooth transition from condyle cartilage to the subchondral bone physiologically[41]. However, its microstructural, micromechanical, nanocompositional, and biomolecular features change remarkably in OA[42]. Consequently, we checked the remodeling of the osteochondral interface by atomic force microscopy and SEM-energy-dispersive X-ray spectroscopy (SEM-EDS) (Fig. 6). The atomic force microscopy results showed that the Young's modulus of the osteochondral interface increased gradually, from $0.72 \pm 0.22$ MPa to $9.06 \pm 0.53$ MPa in the

Control condyles. However, the gradation in tissue modulus was misarranged in UAC group, with a stiffer matrix (up to 17.04 MPa), which was restored after treatment with OSPPB but not Celecoxib (Fig. 6a). As illustrated in Fig. 6b, the Ca and P distributions were notably extended into the condylar cartilage in the UAC group and UAC+Celecoxib group. By contrast, condyles treated with the OSPPB hydrogel displayed Ca and P elemental mappings that closely mimicked those of the Control group. We distinguished the condylar cartilage from subchondral bone based on backscattered electron imaging (Supplementary Fig. 10), because of their differences on inorganic and organic components[42]. Neurovascularization at the osteochondral interface ultimately leads to alterations in its microstructure and micromechanical properties, which are intimately associated with the progression of arthritis[43,44]. We substantiated that the OSPPB hydrogel could alleviate the microstructural and micromechanical changes induced by TMJOA.

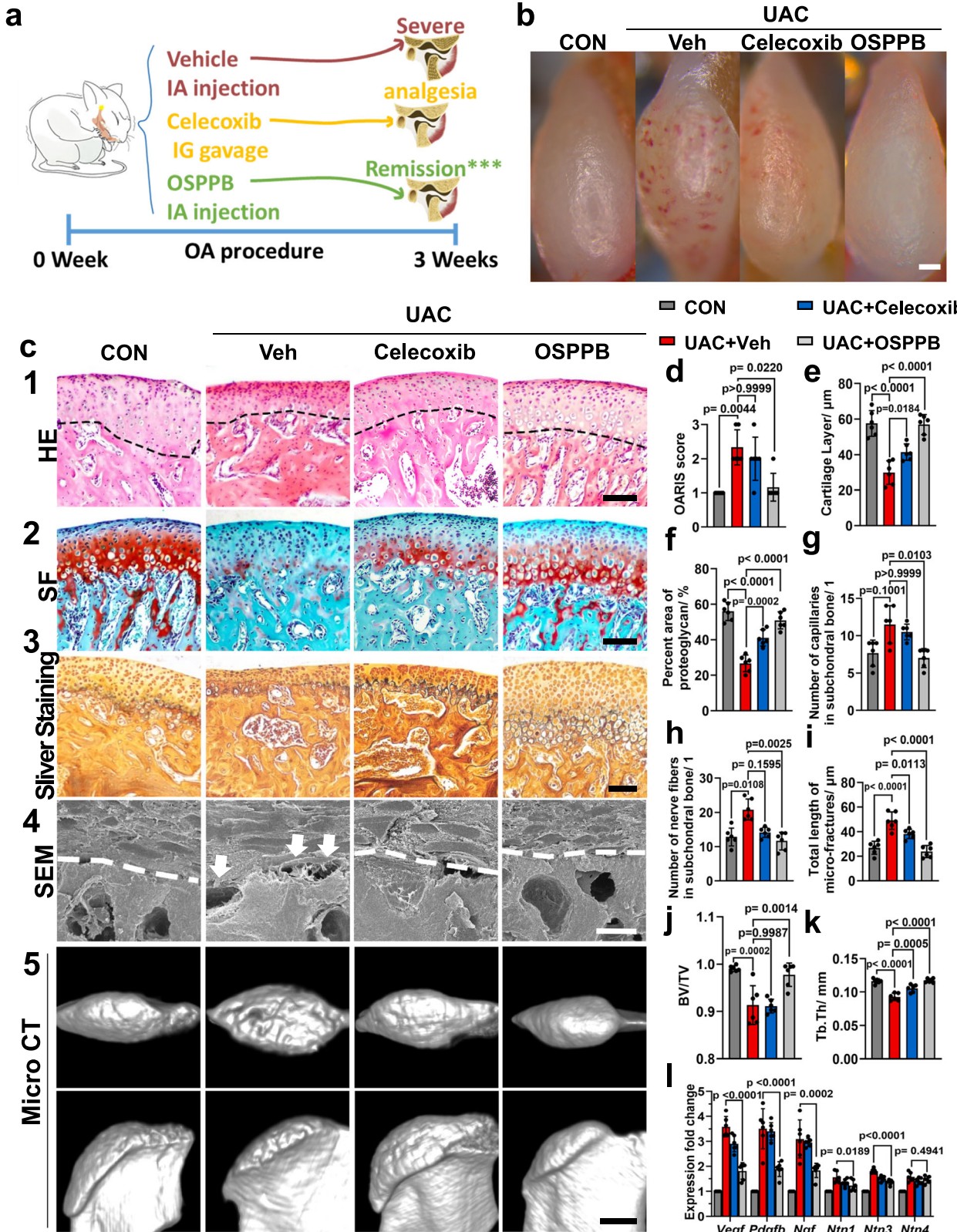

### Effect of the OSPPB hydrogel on osteoarthritic pain in vivo

Through histological and microstructural characterizations, the restorative effect of OSPPB on TMJOA was confirmed; however, whether this treatment could alleviate osteoarthritic pain remained unknown. Pain cannot be measured directly in rodents; therefore, "pain-like behavior" was tested. Spontaneous behavioral responses (the Elevated Plus Maze (EPM) and Open Field Test (OFT)) and evoked behavioral responses (von Frey and electroencephalography) were assessed. Classically, the EPM and OFT were developed to evaluate anxiety in rodents[45]. Given that persistent pain can eventually lead to anxiety[46], we employed these two behavioral tests before sacrifice to indirectly decipher the analgesic effectiveness of the OSPPB hydrogel and Celecoxib on TMJOA-induced pain. Increased open arm entries, the open arm time, and total arm entries in EPM substantiated the

**Fig. 5 | The OSPPB hydrogel alleviates TMJOA in mice. a** The facile fabrication route depicting the design and the results of the in vivo experiment. Reproduced under the terms of a Creative Commons Attribution license (CC-BY-4.0)[20]. **b** Illustrative macroscopic views above the condyles at 3 weeks after intra-articular injection, depicting comparative morphology in the control (CON) and experimental (unilateral anterior crossbite, UAC) groups. **c** Representative images of H&E staining, SF staining, sliver staining, SEM (arrows indicate micro-fractures), and micro-CT. **d**–**k** OARSI score and semi-statistical analysis of the images in (**c**). $n = 6$, $H(3) = 16.12$, $p = 0.0011$ in (**d**). $n = 6$, $F(3, 20) = 28.63$, $p < 0.0001$ in (**e**). $n = 6$, $F(3,$

$20) = 43.48$, $p < 0.0001$ in (**f**). $n = 6$, $H(3) = 14.56$, $p = 0.0022$ in (**g**). $n = 6$, $H(3) = 14.92$, $p = 0.0019$ in (**h**). $n = 6$, $F(3, 20) = 27.57$, $p < 0.0001$ in (**i**). $n = 6$, $F(3, 20) = 16.42$, $p < 0.0001$ in (**j**). $n = 6$, $F(3, 20) = 38.22$, $p < 0.0001$ in (**k**). **l** qRT-PCR analysis of the gene expression of neurovascular factors (*Vegf*, *Pdgfb*, *Ngf*, *Ntn1*, *Ntn3*, *Ntn4*) in the condyles. Scale bars = 200 μm (**b**), 100 μm (**c1**–**3**), 10 μm (**c4**), and 500 μm (**c5**). Data are shown as the means and standard deviations. Kruskal–Wallis (KW) test is followed by Dunn's test for post hoc multiple comparisons in (**d**, **g**, **h**). One-way ANOVA is followed by the Tukey-Kramer method for post hoc multiple comparisons in (**e**, **f**, **i**–**l**).

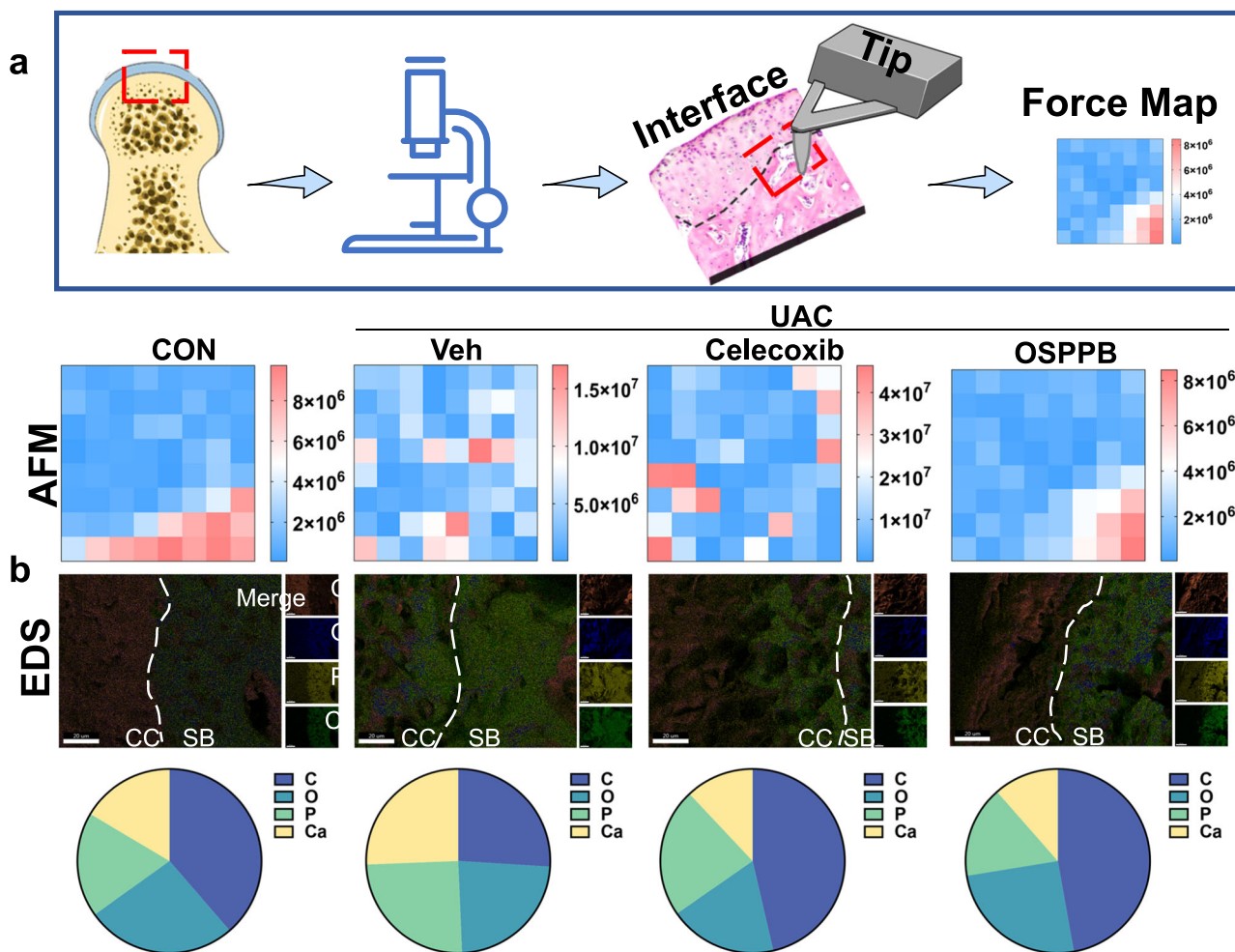

**Fig. 6 | The OSPPB hydrogel ameliorates changes at the osteochondral interface caused by TMJOA. a** Mechanical characterization of the TMJ osteochondral interface by AFM at 3 weeks after different treatments in the CON and UAC groups. **b** SEM-EDS micrographs of osteochondral interface. Scale bars = 20 μm.

alleviation of anxiety-like behavior by the Celecoxib and OSPPB hydrogel (Fig. 7a, c–e). The results from the OFT were consistent with this observation (Fig. 7b, f–h). Based on these observations, it was speculated that the Celecoxib and OSPPB hydrogel administration might relieve the persistent pain-induced anxiety. To evaluate the maxillofacial pain intuitively, mechanical allodynia caused by the von Frey fiber was tested. The withdrawal threshold decreased significantly in UAC + Veh group, but returned to control levels after treatment with the Celecoxib and OSPPB hydrogel (Fig. 7i). In addition, an electroencephalogram of the primary sensory cortex S1BF[47] was captured immediately following small brush stimulation (Fig. 7j). The low timestamp precision meant that some interference occurred around the intended timestamps. Nonetheless, we observed lower spectral energy after Celecoxib and OSPPB treatment in the low frequency band (0–20 Hz), which was similar to that of the Control group. These

preliminary results indicated that mice exhibited significant alleviation of maxillofacial pain after receiving Celecoxib and OSPPB hydrogel treatment.

Additional experiments were performed to validate the significant anatomical changes related to pain relief. The co-staining of SYTO RNASelect Green Fluorescent Cell Stain (an RNA-specific stain) and E-cadherin (cytomembrane) was used to identify exRNA at the osteochondral interface. The extracellular flocculently-arranged RNA far away from E-cadherin-positive area was defined as exRNA in immunofluorescence staining[11]. As shown in Fig. 8a, d, the amount of exRNA decreased after OSPPB hydrogel but not Celecoxib treatment. The anterograde tracing exhibited a reduced distribution of trigeminal ganglion derived-axon endings in the UAC + OSPPB group (Fig. 8b, e). Besides, the results from confocal laser scanning microscopy of protein gene product 9.5 (PGP 9.5) and calcitonin gene-related peptide

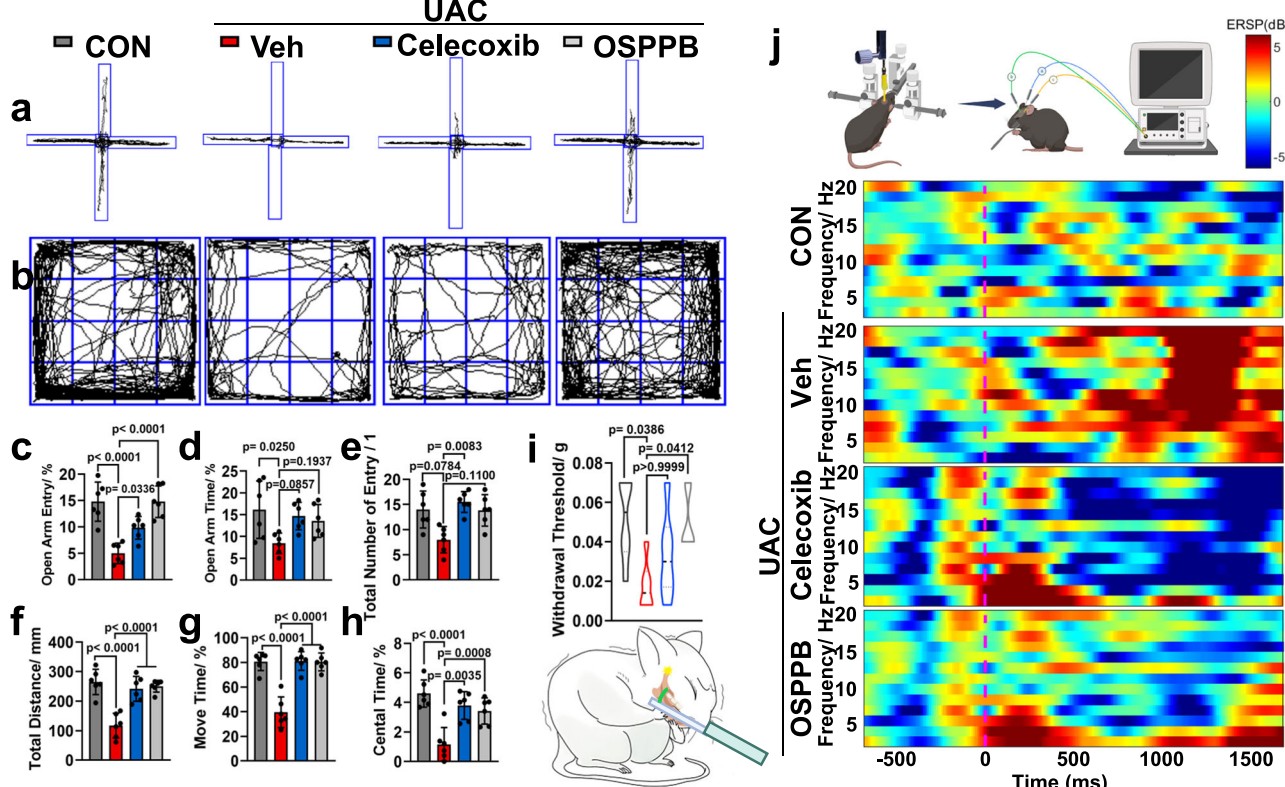

**Fig. 7 | Celecoxib and OSPPB hydrogel exhibit remarkable pain-alleviating functionality. a** Representative tracks of mice at 3 weeks in the CON and UAC groups, as documented in the EPM test, with the corresponding statistical results (**c**–**e**). One-way ANOVA is followed by the Tukey-Kramer method for post hoc multiple comparisons in (**c**, **d**). $n = 6$, $F_{(3, 20)} = 16.83$, $p < 0.0001$ in (**c**). $n = 6$, $F_{(3, 20)} = 3.696$, $p = 0.0289$ in (**d**). $n = 6$, $H(3) = 11.6$, $p = 0.0089$ in (**e**). **b** Representative tracks of mice in the OFT, with the corresponding statistical results (**f**–**h**). One-way ANOVA accompanied by the Tukey-Kramer method is used for post hoc multiple comparisons. $n = 6$, $F_{(3, 20)} = 19.4$, $p < 0.0001$ in (**f**). $n = 6$, $F_{(3, 20)} = 31.85$,

$p < 0.0001$ in (**g**). $n = 6$, $F_{(3, 20)} = 13.68$, $p < 0.0001$ in (**h**). **i** Results of the von Frey test revealing the pain relief after different treatments ($n = 6$, $H(3) = 10.21$, $p = 0.0168$). Reproduced under the terms of a Creative Commons Attribution license (CC-BY-4.0)[20]. **j** EEG spectrograms of the stimulus before and after small brush stimulation. Created in BioRender. Data are shown as the means and standard deviations. One-way ANOVA is followed by the Tukey-Kramer method for post hoc multiple comparisons in (**c**, **d**, **f**–**h**). Kruskal–Wallis (KW) test is followed by Dunn's test for post hoc multiple comparisons in (**e**, **i**).

(CGRP) were consistent with the aforementioned findings (Fig. 8b, f, g). Also, compared with UAC + Celecoxib group, the platelet and endothelial cell adhesion molecule 1 (PECAM1, also known as CD31) positive vessels and the neurovascular factor VEGF levels diminished after OSPPB hydrogel treatment (Fig. 8c, h). The expression of several factors closely related to skeletal interoception and pain[48], including Cyclooxygenase 2, DCC netrin 1 receptor (DCC), and substance P (SP), were further scrutinized, and the results revealed the reduction in pain mediators after OSPPB hydrogel treatment (Fig. 8c, i–k). Taken altogether, these data indicated that Celecoxib demonstrated therapeutic efficacy primarily in pain alleviation, yet its influence on regenerative microenvironment was found to be substantially limited. However, the OSPPB hydrogel effectively hampered neurovascularization at the osteochondral interface and alleviated osteoarthritic pain.

## Discussion

In this study, we fabricated a dual-responsive OSPPB hydrogel to inhibit neurovascularization at the osteochondral interface in OA. This multifunctional hydrogel comprised an OSAP polymer, a PPCA polymer, and BGN@Be. The OSPPB hydrogel was formed through a reversible Schiff base bonds and boronic ester bonds between OSAP and PPCA, and $Ca^{2+}$ coordination between BGN and OSAP. The presence of the Schiff base bonds and borate ester bonds meant that the OSPPB hydrogel exhibited self-healing, injectability, dual-responsiveness, and could realize sequential and on-demand delivery of drugs. It scavenged exRNA via polycationic PPCA and neutralized the function

of VEGF via BGN@Be. Consequently, it not only directly suppressed the activity of neurovascular factors, but also interfered with their recruitment process, ultimately resulting in significant inhibition of neurovascularization at the osteochondral interface.

This study builds on the previous mechanistic research and extends into the exploration of biomaterials. The previous data have shown that exRNAs have the ability to recruit polycationic neurovascular factors, which further amplified abnormal neurovascularization in the osteoarthritic condylar joint and resulted in unbearable pain[11]. Besides, given that the osteochondral microenvironment in OA exhibits elevated levels of ROS and a decreased pH[31], we have designed this unique material to achieve precise intervention in the neurovascularization in OA. This article follows a logical sequence from material synthesis, through in vitro experiments, to in vivo testing, systematically validating the efficacy of the hydrogel. Firstly, we fabricated OSPPB hydrogel, following synthesization and characterization of three functional components, OSAP, PPCA and BGN@Be. To validate its gelation, injectability, self-healing, ROS-pH responsiveness, antioxidant properties, positive charge, and biocompatibility, we employed various methods and presented the results in Figs. 1 and 2. Secondly, we demonstrated in vitro that the hydrogel inhibits the function of EPC and TG cells, confirming that the primary active components are PPCA and BGN@Be. Thirdly, we compared the OSPPB hydrogel with the commonly-used celecoxib in in vivo experiments, and confirmed its effectiveness in inhibiting OA disease progression and alleviating OA pain. This research further supports the mechanism

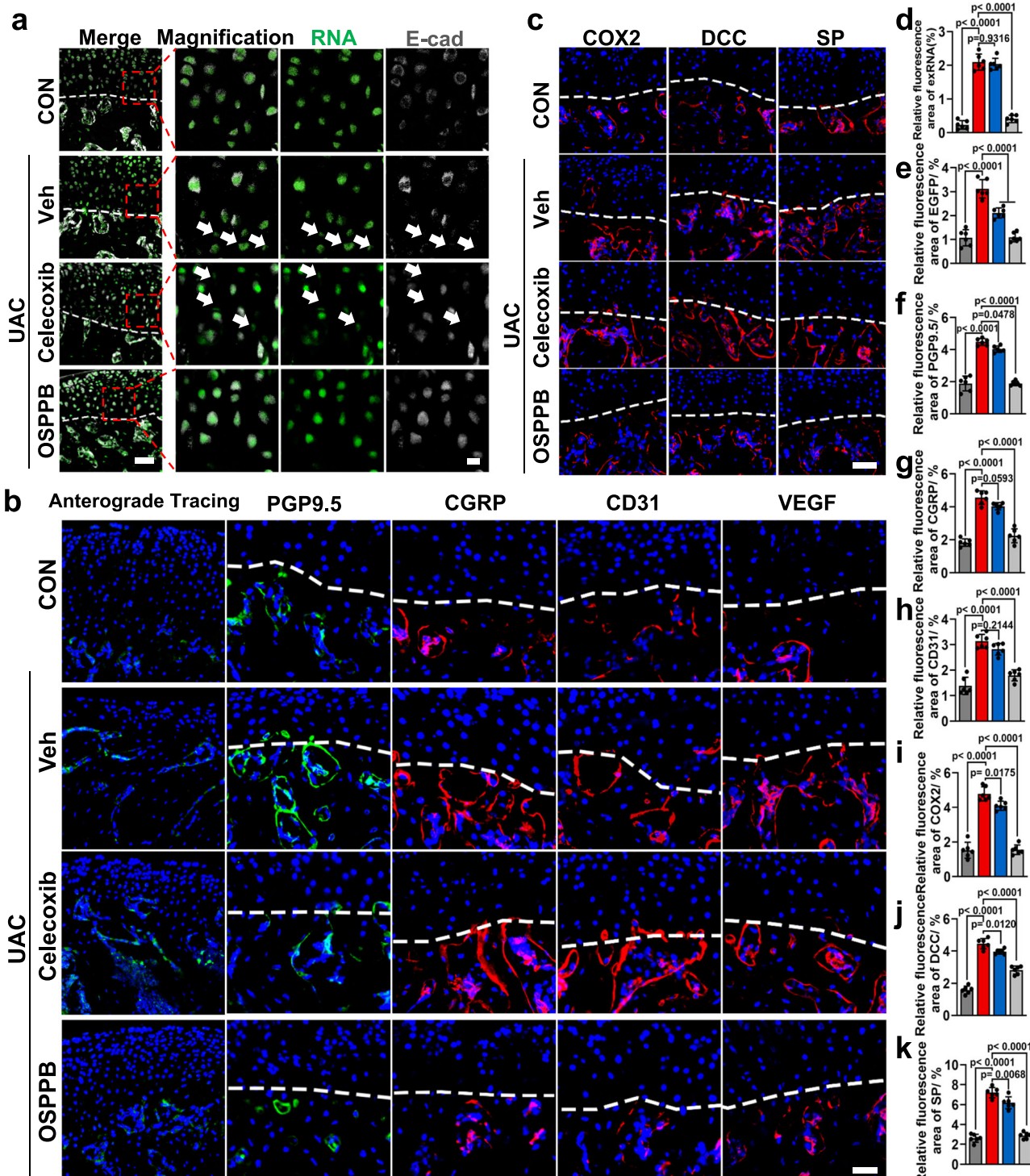

**Fig. 8 | The OSPPB hydrogel effectively attenuates neurovascularization at the osteochondral interface. a** Representative images of the distribution of exRNA at the osteochondral interface of condyles in the CON and UAC groups at 3 weeks after different treatments. **b** Representative images of the trigeminal ganglion anterograde tracing and immunofluorescence staining of nerves (PGP9.5 and CGRP), vessels (CD31), and VEGF. **c** Representative images of the distribution of pain-related factors (COX2, DCC, and SP). **d** The semi-statistical analysis of (**a**). **e**–**h** The semi-statistical analysis of (**b**). **i**–**k** The semi-statistical analysis of (**c**). $n = 6$, $F(3, 20) = 213.6$, $p < 0.0001$ in (**d**). $n = 6$, $F(3, 20) = 66.07$, $p < 0.0001$ in (**e**). $n = 6$, $F(3, 20) = 146.4$, $p < 0.0001$ in (**f**). $n = 6$, $F(3, 20) = 96.29$, $p < 0.0001$ in (**g**). $n = 6$, $F(3, 20) = 57.17$, $p < 0.0001$ in (**h**). $n = 6$, $F(3, 20) = 130.5$, $p < 0.0001$ in (**i**). $n = 6$, $F(3, 20) = 159.9$, $p < 0.0001$ in (**j**). $n = 6$, $F(3, 20) = 143.9$, $p < 0.0001$ in (**k**). Scale bars = 50 μm (left in **a**), 10 μm (right in **a**), 30 μm (**b**, **c**). Data are shown as the means and standard deviations. One-way ANOVA is followed by the Tukey-Kramer method for post hoc multiple comparisons.

that exRNA promotes neurovascularization and leads to OA disease progression, and provides a new therapeutic approach and offers insights into the development of inorganic-organic composite materials.

There is a growing consensus that neurovascularization at the osteochondral interface represents a pivotal pathological change underpinning the chronic pain and progressive articular degradation in OA. Consequently, mechanistic research and therapeutic

interventions targeting this pathological change in OA have gained increasing attention. On the one hand, mechanistic research has primarily focused on neurovascular factors. Research has demonstrated that the knockout of *Ntn1* (encoding Netrin1) in tartrate-resistant acid phosphatase positive osteoclasts effectively inhibited aberrant subchondral bone remodeling, sensory nerve sprouting, and reduced osteoarthritic pain behavior[49]. Moreover, nerve growth factor directly activated nociceptive sensory nerves and induced immune activation, thereby impacting the subchondral bone environment and leading to osteoarthritic pain[50]. Notably, anti-nerve growth factor antibodies preliminarily exerted anti-inflammatory properties and pain control capabilities; however, they are associated with certain side effects and safety concerns[50]. Ma established the distinct roles of VEGF in promoting joint pathogenesis via VEGF Receptor 1 and exacerbating joint-related nociception via VEGF Receptor 2[51], and innovatively utilized nanotechnology-based particles for OA treatment[52]. On the other hand, there is also compelling evidence underscoring the importance of the recruitment of neurovascular factors. Polyanionic exRNA could facilitate the mobilization of positively charged neurovascular factors, thereby contributing to the induction of neurovascularization[11]. Moreover, exRNA could also directly induce angiogenesis, which was inhibited by RNase[15,53]. The present study fabricated an OSPPB hydrogel incorporating positively charged PPCA and bevacizumab-loaded BGN, which was capable of concurrently scavenging exRNA and blocking VEGF. The in vitro findings revealed a pronounced functional suppression of both TG cells and EPCs. Celecoxib, a nonsteroidal anti-inflammatory drug widely employed in OA management, served as first-line treatment in in vivo studies[5]. Results indicated that celecoxib efficaciously alleviated pain yet exerted minimal impact on optimizing the OA microenvironment, including OARSI scores, neurovascular patterns and bone density. Conversely, OSPPB hydrogel treatment matched celecoxib's analgesic efficacy and demonstrably ameliorated OA conditions. This underscored OSPPB intervention manifested remarkable efficacy in inhibiting neurovascularization, suppressing joint destruction, and ameliorating the persistent pain associated with OA.

Compared with Ma's research, which used nano-particles to deliver anti-VEGF antibodies[52], our study utilized a hydrogel incorporated with bevacizumab-loaded BGN, with enhanced biocompatibility and an on-demand release ability. Over recent decades, the strategy of treating OA has evolved from mono-component, mono-function to multi-component, multi-function strategies[54]. The responsiveness and targetability of hydrogels are essential attributes to optimize precise drug delivery and minimize off-target effects[31]. The OSPPB hydrogel incorporated Schiff base bonds and boronic ester bonds, endowing it with superior sensitivity to high ROS levels and acidic environments, thus allowing for stimuli-responsiveness and the controlled release of bioactive payloads. Moreover, the incorporation of positively charged PPCA allowed accumulation at the osteochondral interface through passive targeting[31]. The above two characteristics (responsiveness and specificity) determined the applicability of the hydrogel, whereas the embedded active ingredients fundamentally dictated its overall effectiveness. Currently, hydrogels mainly contribute to OA treatment through three principal modalities: lubrication, drug delivery, and tissue repair[54]. From the aspect of lubrication, hydrogels typically enhance lubrication by forming a viscoelastic hydration layer on the cartilage surface through hydrogen bonding, coordination interactions, and electrostatic interactions. In this study, we selected sodium alginate, a widely recognized and naturally sourced polysaccharide extracted from marine algae, as the foundational component, leveraging its established lubricating properties[55]. Regarding drug delivery, a hydrogel should exhibit a remarkable capacity for sustained and efficient drug release, to mitigate potential adverse effects often associated with immediate release. Specifically, the OSPPB hydrogel realized the responsive release of the active constituents, including PPCA and BGN@Be. The later one possessed a controllable nanostructure, which allowed for the effective slow release of the loaded drug, bevacizumab. In the context of tissue repair, the endeavor to fully reconstruct and reconstitute the intricate architecture and physiological functionality of articular cartilage and osteochondral tissues remains a significant scientific challenge. In this study, we employed BGN@Be, characterized by their excellent biodegradability, straightforward synthesis process, and effective tissue repair capabilities[56,57]. BGN@Be realized that its early release of bevacizumab effectively inhibited pathological neurovascularization, followed by a subsequent slow release of bioactive ions, which has been proven to facilitate tissue repair[56,57]. This dual functionality ensured the structural restoration and functional rehabilitation of arthritic tissues, thereby embodying a sophisticated approach to therapeutic intervention. These key functional aspects have been meticulously considered in the OSPPB hydrogel, further substantiating its superiority in addressing OA treatment challenges.

Despite the auspicious therapeutic outcomes observed in TMJOA treatment using the OSPPB hydrogel, several limitations require explanation and further refinement. Firstly, the therapeutic effectiveness of the OSPPB hydrogel was only substantiated using murine OA models; thus, the validity of these findings necessitates further exploration in larger animal models. Secondly, the present study only investigated the suppressive capability of the OSPPB hydrogel concerning vessels and sensory nerves; however, its potential impact on sympathetic nerves is unknown. This is particularly pertinent because of the function of sympathetic nerves in skeletal interoception, and the well-established anatomical and functional interdependence between sympathetic nerves and the vascular network[48,58,59]. Thirdly, the synthesis and storage of hydrogels incorporating Schiff base bonds and boronic ester bonds requires a sophisticated methodology, which might limit the large-scale reproducibility and widespread practical deployment of such hydrogels. Fourthly, the real-time monitoring of ROS and pH changes within the joint, as well as the real-time degradation of the hydrogel, has not yet been achieved. This will require further investigation in future studies. Additionally, the reactivity of the degradation products with other cells in the OA microenvironment also warrants further investigation. For example, whether the degraded hydrogel components are phagocytosed by macrophages and lead to their polarization, and how this affects OA tissue repair and pain[60], remains to be explored. Nevertheless, this newly developed OSPPB hydrogel, a kind of inorganic-organic composite materials, might also have the potential to treat diseases characterized by aberrant neurovascularization, such as tumors and heterotopic ossification, which requires further exploration and validation.

In conclusion, we synthesized an injectable, self-healing, pH/ROS dual-responsive, and polycationic hydrogel that reversed osteoarthritic changes and alleviated chronic pain by inhibiting neurovascularization at the osteochondral interface. The positive charged OSPPB hydrogel overcomes the "joint barrier", penetrates the cartilage matrix, and functions at the osteochondral interface. Stimulated by low pH and high ROS, the dynamic Schiff base bonds and boronic ester bonds rapidly cleave, leading to degradation of the hydrogel network, exposure of positively charged PPCA polymers, and the detachment of BGN@Be. Furthermore, exRNA scavenging and controlled release of bevacizumab directly inhibit neurovascularization, relieve pain, and restore the anatomical changes caused by OA. Although the results are preliminary, the reported OSPPB hydrogel-based strategy demonstrates noteworthy therapeutic efficacy in addressing TMJOA, thereby paving the way for development of an inspirational panacea for OA in other joints.

## Methods

### Synthesis of OSPPB hydrogels

**Preparation of OSAP.** One gram of sodium alginate (S278630, Aladdin, Shanghai, China) was dissolved in 100 mL deionized water. After it

was completely dissolved, 0.985 g of 1-(3-Dimethylaminopropyl)-3-ethylcarbodiimide hydrochloride (EDC·HCl, E8170, Solarbio, Beijing, China) was added. Fifteen minutes later, 0.49 g of 3-aminophenyl boronic acid (924879, Jkchemical, Beijing, China) was added. The mixture was stirred for 24 h, and then dialyzed against water (molecular weight cut-off: 3500 Da, changed every 8 h) for 7 days at room temperature to remove the unreacted substance. After lyophilization, sodium alginate phenylboronic acid (SAP) was obtained. Then, 1 g of SAP was dissolved in 100 mL of deionized water for 2–3 h. Five milliliters of deionized water dispersed with 210 mg of $NaIO_4$ (S104093, Aladdin) was dripped into the SAP solution in the dark. Two and a half hours later, 1 mL of ethylene glycol (E808735, Macklin, Shanghai, China) was added and incubated for 1 h to neutralize the unreacted $NaIO_4$. Dialysis and lyophilization were conducted as mentioned above to obtain OSAP. Until further use, SAP and OSAP were stored at 4 °C. The characterization of SAP and OSAP were conducted using ATR-FTIR.

**Preparation of PPCA.** Polyethyleneimine (4.5 g, E808879, Macklin) was dissolved in 20 mL of MES buffer solution (M301885, Macklin). Protocatechuic acid (308.24 mg, 367266, Jkchemical) and EDC·HCl (1.54 g) were dissolved in 80 mL of MES buffer solution under ice bath conditions for 0.5 h. Then, 920.72 mg of N-Hydroxysuccinimide (6066-82-6, Solarbio) was added and stirred for 12 h to active the carboxyl groups in the dark. This solution was then added into the previously prepared polyethyleneimine solution. After the reaction was stirred at room temperature in the dark for 24 h, dialysis was conducted for 7 days (molecular weight cut-off: 1800 Da, changed every 8 h). The dialysate was lyophilized to obtain PPCA and PPCA was stored at 4 °C for further analysis. The characterization of PPCA was conducted using ATR-FTIR.

**Preparation of BGN@Be.** The synthesis of BGN was performed according to Niu et al.[57], with a slight adjustment. In brief, the BGN nanoparticles, comprising $SiO_2$-CaO-$P_2O_5$ (mol.%) = 80:16:4, were synthesized by a sol-gel method and obtained by freeze-drying and high temperature calcination. Then, 20 mg of BGN was dispersed in 20 mL of PBS using ultrasonic treatment. Bevacizumab (50 μg; A2006, Selleck, Shanghai, China) was added, the mixture was stirred for 24 h in an ice bath, and centrifuged (4 °C, 11,433 × g, 15 min) to obtain the precipitate, and then washed with PBS three times. The BGN@Be powder was obtained by lyophilization and stored at 4 °C until further use. The morphology, size distribution, and phase structure were investigated using TEM, nanoparticle tracking analysis, and XRD, respectively. Degradation was detected using inductively coupled plasma-mass spectrometry (Agilent 7800, Agilent Technologies, Santa Clara, USA) and ultraviolet-visual (UV−Vis) spectrophotometry. The supernatant obtained after centrifugation (4 °C, 11,433 × g, 15 min) was processed daily and analyzed according to the instrument's instructions.

**Synthesis of OSPPB hydrogels.** For OSPPB hydrogel preparation, BGN@Be was dispersed in PBS at 1 mg/mL to obtain Solution A. Solution B was created by dissolving 100 mg of OSAP in 2 mL of Solution A. Solution C was prepared by dissolving 100 mg of PPCA in 2 mL Solution A. Then Solution B and Solution C were thoroughly blended to obtain the OSPPB hydrogel.

### Physiochemical characterization of the OSPPB hydrogels
**Morphological characterization.** To characterize the morphology of the hydrogel, the specimens were lyophilized, sputter-coated with gold, and examined by field-emission scanning electron microscopy (FE-SEM, S-4800, Hitachi, Tokyo, Japan) at a 5 kV accelerating voltage. Additionally, energy-dispersive X-ray spectroscopy (Element EDS System, Ametek, PA, USA) was used to characterize the mineral elemental composition and BGN@Be distribution in the OSPPB hydrogels.

**ATR-FTIR.** Specimens were scanned by ATR-FTIR (FTIR-8400S, Shimadzu, Tokyo, Japan) from 4000 to 400 cm$^{-1}$, with 32 scans averaged at a resolution of 4 cm$^{-1}$. Spectral analysis was performed using IR solution software (Shimadzu, Kyoto, Japan).

**Rheology measurements.** A rheometer (NETZSCH Kinexus Lab+, Bayern, Germany) was used to measure the rheological properties of the OSPP and OSPPB hydrogels using 3 mL hydrogel for each test, which was carried out at 25 °C. The time sweep sequence of the OSPP and OSPPB hydrogels was performed at a fixed frequency (1 Hz) and fixed strain (1%). Besides, the frequency sweep sequence was conducted between 0.1 and 10 Hz under a fixed strain (2%). In the shear-thinning test, the shear rate ranged from 0 to 100 s$^{-1}$, and the frequency and strain were maintained at 10 rad s$^{-1}$ and 1%, respectively.

**In vitro degradation of the hydrogel.** Degradation of the OSPP and OSPPB hydrogels in vitro was investigated in PBS at 37 °C. Hydrogel samples were immersed by PBS after weighting and recording (M0, mg). Then, samples were taken out at specific times (1, 2, 3, 4, 5, and 6 days), dried with filter paper, weighed, and recorded (Mt, mg). The degradation ratio was calculated: Remaining hydrogel (%) = (Mt/M0) × 100%.

**Response of the OSPPB hydrogel to pH and ROS change.** Twenty microliters of HCl (pH = 5) were used to adjust the pH of the medium and apply pH stimulation. In addition, 20 μL of $H_2O_2$ (1 mM) was used to increase the ROS of the medium and apply ROS stimulation. The OSPPB hydrogel stimulated using 1 mL of PBS + HCl + $H_2O_2$ was placed at 37 °C for 4 h before photographing. Then, the stimulated hydrogels were conducted rheology measurements using rheometer (NETZSCH Kinexus Lab+, Bayern, Germany). The frequency sweep sequence was conducted between 0.1 and 10 Hz under a fixed strain (2%). After lyophilizing the samples, we used SEM for morphological examination.

**Anti-inflammatory evaluation.** We conducted a DPPH radical scavenging activity assay. Aqueous extracts of different concentrations were obtained by complete dissolution of the OSPPB hydrogel in PBS. Hydrogel extracts of different concentrations were dispersed in 3 mL of DPPH methanol solution (0.1 mg/mL). The DPPH methanol solution without materials and Vitamin C were chosen as antioxidant negative and positive controls, respectively. After photography, the mixture was shaken vigorously and allowed to stand at 37 °C for 30 min. The absorbance of the mixture was then measured at 528 nm.

**Zeta potential.** The OSPP and OSPPB extracts were tested initially for transmittance. The zeta potential was tested when the transmittance reached 80% using a Litesizer 500 particle analyzer (Anton Paar, Graz, Austria).

**Binding test.** VEGF pre-coated microtiter plate wells were obtained and blocked using Tris-buffered saline supplemented with 3% bovine serum albumin (821006, MilliporeSigma, Burlington, MA, USA). For group 1, Cy3-RNA (50 μL, 150 μg/mL, CY3-AAAAAAAAAAAAAAAAAAA AAAAAAAAAAAAAAAAAAAAAAAAAAAAAAAAA, Sangon, Shanghai, China) and diethyl pyrocarbonate water (50 μL) were incubated in the wells at 22 °C for 2 h. For group 2, 50 μL of the OSPPB hydrogel replaced the diethyl pyrocarbonate water. For group 3, the wells obtained in group 1 were then incubated with 50 μL of the OSPPB hydrogel at 22 °C for another 2 h. The wells were then observed using CLSM (Nikon A1R, Nikon Corporation, Minato-ku, Tokyo, Japan), in which the residual Cy3 fluorescence referred to the bound RNA[11].

**Molecular dynamics (MD) simulations.** RNA structure was modeled based on our previous work[11]. Initial coordinates of VEGF were taken from the global Protein Data Bank (PDB: 1VPF). The PEI was modeled

using Charmm-gui (https://charmm-gui.org). Ten VEGF and ten PEI and a 50nt RNA were placed into a 260 × 242 × 235 Å3 rectangle box using packmol. Then 448805 TIP3P water molecules and 99 Na$^+$ were placed into the box using tleap module of AmberTools. The ff14SB, OL3, GAFF2 force field was used for the protein, RNA and PEI, respectively. The solvated system were energy minimized with 5000 step steepest-descent and 5000 step conjugate gradient algorithms. The system was then heated from 0 K to 300 K over 500 ps. Next, the system was further relaxed by restrained MD simulation with the force constant of 5.0 kcal/(mol Å2) in the canonical ensemble (NVT) ensemble and then constant number, pressure and temperature (NPT) ensemble for 500 ps for each until the energy, pressure and temperature reached equilibrium. T. Finally, 200 ns MD simulation was performed in the NPT ensemble (constant number of atoms, pressure, and temperature). The binding free energy is calculated by MM-GBSA method.

**Hemolysis evaluation.** One milliliter of heparinized mouse blood was suspended in 9 mL of PBS, and centrifuged to separate the red blood cells (257 × $g$, 5 min). Then, we repeated this process three times until the supernatant became clear. The RBC pellet was incubated with extracts at different concentrations for 1 h at 37 °C, followed by centrifugation (714 × $g$, 5 min, 4 °C). PBS and Triton X-100 (112298, MilliporeSigma) were employed as the negative and positive controls, respectively. The photos were taken, and the absorbance of the supernatant was measured 540 nm.

## In vitro experiments

**EPC culture.** We cultured and treated EPCs according to the previous study[11]. After sterilization with 75% ethanol for 24 h, the hydrogel was washed with PBS thoroughly to eliminate residual ethanol. The EPCs (Jennio Biotech, Guangdong, China) were obtained and the control medium was prepared by adding 1% penicillin/streptomycin (516106-20MLCN, MilliporeSigma) and 10% fetal bovine serum (Gibco, Thermo Fischer Scientific, Waltham, MA, USA) to Dulbecco's modified Eagle's medium (Hyclone, Logan, UT, USA). The EPCs were cultured at 37 °C in 5% $CO_2$ on the hydrogel at a density of $5 \times 10^4$ cells/mL, changing the medium every other day. The EPCs were divided into different groups: No treatment, VEGF (2.5 ng/mL), RNA-VEGF complex (2.5 ng/mL), and RNA-VEGF (2.5 ng/mL)-OSAP, RNA-VEGF (2.5 ng/mL)-PPCA, RNA-VEGF (2.5 ng/mL)-BGN@Be, and RNA-VEGF (2.5 ng/mL)-OSPPB. The EPCs were analyzed using a cell counting kit-8 assay (Dojindo, Tokyo, Japan) and live/dead cell staining according to the kit's instructions. For the cell scratch test, EPCs in monolayers were cultured in different conditioned media for 24 h after scratching. For the Transwell cell migration assay, the underside of the Transwell inserts was stained with crystal violet after 24 h of incubation. For the tube formation assay, the Matrigel matrix (Corning Inc., Corning, NY, USA) were used according to manufacturer's protocols. ImageJ software (National Institute of Health, Bethesda, MD, USA) was used to analyze the data.

**TG neuron culture.** The TG cells were isolated from neonatal Sprague-Dawley rats (1–5 days old) using a previously published protocol[11]. In brief, after decapitation, the trigeminal ganglia were aseptically dissected and digested in an enzymatic cocktail (0.1% collagenase and 0.25% trypsin, Thermo Fisher Scientific). Following centrifugation, the pellet was resuspended and plated on poly-d-lysine hydrobromide (50 ng/mL, Thermo Fisher Scientific)-coated cell culture plates in Neurobasal media (Invitrogen, Waltham, MA, USA) containing 0.5 mM L-glutamine, 1% penicillin-streptomycin, and 2% B-27 (Gibco). The cells were incubated in 5% $CO_2$ at 37 °C, changing the medium every other day. Four hours later, the TG neurons were divided into seven groups as mentioned in section "EPC culture". The immunofluorescence staining of β3-tubulin (1:300, ab18207, Abcam, Cambridge, UK) and crystal violet were used to reveal its axonal morphology.

## Animal experiments

**Animals and ethics statement.** Forty-eight female C57BL/6J mice (8 weeks old, 17–19 g) and thirty Sprague-Dawley rats (1–5 days old) were purchased from the Laboratory Animal Center of the Forth Military Medical University. Animals were housed in a controlled environment set at 24 °C and 55% humidity, with a 12-h light/dark cycle. Females show a high morbidity rate of TMJOA[61]; therefore, only female animals were used. The animal protocols were approved by the Institutional Animal Care and Use Committee of the Forth Military Medical University (IACUC-20240011) and were caried out according to the "Animal Research: Reporting of In Vivo Experiments" guidelines for preclinical animal studies.

**Murine OA model and in vivo injection.** Mice were randomly divided into four groups: the sham-operated control group (CON), the unilateral anterior crossbite + vehicle group (UAC + Veh, Veh meant PBS), the unilateral anterior crossbite + Celecoxib gavage (UAC + Celecoxib) and the unilateral anterior crossbite + OSPPB injection group (UAC + OSPPB). Mice treated with UAC were anesthetized with 1% intraperitoneal sodium pentobarbital and the UAC procedure was applied by adhering a metal tube to the left incisor[34–36]. Mice in the CON group went through a similar procedure, but without metal tube fixation.

For TMJ injection, mice were anesthetized and placed on one side. A needle with a custom-designed Hamilton syringe was inserted from the zygomatic arch between the corners of the eye and ear, along the bony wall, to finally reach the TMJ. Vehicle (PBS, 20 µL) and OSPPB hydrogel (20 µL) were injected into UAC + Veh group and UAC + OSPPB group, respectively. The injection was carried out immediately after the UAC device was pasted. The mice in UAC + Celecoxib group were simultaneously orally administered with celecoxib (80 mg/kg). The dosage of 80 mg kg$^{-1}$ day$^{-1}$ in mice is equivalent to the dose of 400 mg day$^{-1}$ for a 60 kg person[62].

A stereotaxic apparatus (68018, RWD, Shenzhen, China) was used for anterograde tracing and electroencephalography. For trigeminal ganglion injection and anterograde tracing, rAAV-hSyn-EGFP-WPRE-hGH polyA (BrainVTA, Wuhan, China; $2 \times 10^{12}$ viral genomes/mL), a recombinant self-fluorescent adeno-associated virus, was used. The injection process was the same as previously reported[11]. The coordinates of trigeminal ganglion injection were Medial-Lateral (ML): ±1.45, Anterior-Posterior (AP): −1.34, Dorsal-Ventral (DV): −5.65 (bregma). For implantation of the electroencephalography electrodes, monopolar insulated stainless-steel electrodes (QZY-2, Jingong, Wenzhou, China) were used[38]. The coordinates of the primary sensory cortex S1BF were ML: ±2.6, AP: −1.6, DV: −1.23, and the cerebellum coordinates were ML: 0, AP: −15.5, DV: −2.5. One week after operation, the UAC procedure and TMJ injection were carried out.

Three weeks later, pentobarbital overdose was used to euthanize all the mice. The major organs and condyles were harvested and subsequently subjected to further processing in accordance with the experimental protocol. The major organs were then sectioned, and histologically stained with H&E to detect the toxicity of the hydrogels after 3 weeks.

**In vivo distribution.** Rhodamine B-stained BGN@Be was doped in the OSPPB hydrogel, which was injected into the TMJ. Upon reaching 30 instances of passive mouth opening, the mouse was humanely euthanized by pentobarbital overdose. The condyles were collected, decalcified, dehydrated, sectioned, and observed under CLSM (Nikon A1R, Nikon).

**Characterization of TMJOA anatomical changes.** Some condyles were collected, fixed in 4% paraformaldehyde for 24 h, immersed in 30% sucrose for 3 days, embedded in optimal cutting temperature compound (Leica, Wetzlar, Germany) and stored at −80 °C. The 5 µm-thick central sagittal sections were obtained and stained with H&E,

safranin O/fast green, glycine silver, and immunofluorescence. For immunofluorescence, the primary antibodies used recognized protein gene product 9.5 (PGP 9.5; 1:300, ab8189, Abcam), CGRP (1:400, 14959, Cell Signaling Technology, Danvers, MA, USA), VEGF (1:300, sc-7269, Santa Cruz Biotechnology CA, USA), CD31 (1:300, sc-376764, Santa Cruz Biotechnology, Santa Cruz, CA, USA), E-cadherin (1:300, ab231303, Abcam), cyclooxygenase 2 (COX2; 1:300, sc-376861, Santa Cruz Biotechnology), DCC (1:300, sc-515834, Santa Cruz Biotechnology), and SP (1:300, sc-21715, Santa Cruz Biotechnology). The chemical dye used was SYTO® RNASelect™ Green Fluorescent Stain (S32703, Invitrogen).

For scanning electron microscopy and energy-dispersive X-ray spectroscopy (SEM-EDS), the condyles were fixed in 2.5% glutaraldehyde, followed by ethanol gradient dehydration and immersion in hexamethyldisilane (Electron Microscopy Sciences, PA, USA). After fixation on sample holders, the specimens were imaged using a field-emission scanning electron microscope (FE-SEM, S-4800, Hitachi, Tokyo, Japan) and Energy-dispersive X-ray spectroscopy (Element EDS System, Ametek, PA, USA).

Some condyles were immediately used for morphological observation, followed by micro-CT, and then cut into 500-μm-thick samples for atomic force microscope (AFM) testing. The harvested specimens were scanned using a micro-CT scanner (Inveon, Siemens AG, Munich, Germany) at a resolution of 8 μm. Image acquisition was performed at 80 keV and 500 mA and a $0.3 \times 0.3 \times 0.3$ mm region was analyzed. For the AFM test, AFM-based nanoindentation (Keysight 5500, Keysight Technologies, Santa Rosa, CA, USA) was used to measure the mechanical properties (force constant = 7.6 N/m, a conical tip).

The remain condyles were used for quantitative real-time polymerase chain reaction. Complementary DNA was synthesized and used as the template in the real-time polymerase chain reaction as previously reported[63]. Gene expression levels were estimated using the $2^{-\Delta\Delta Ct}$ method[64] with the *Gapdh* (encoding glyceraldehyde 3-phosphate dehydrogenase) expression level as the internal control. The primer sequences are presented in the Supplementary Table.

**Pain-like behavioral analyses.** Before sacrifice, the mice underwent behavioral analyses and electroencephalography. For the von Frey test, each mouse was placed individually in a metal mesh chamber in a quiet laboratory with a stable temperature ($22 \pm 1$ °C) and allowed to acclimate for at least 30 min before testing. A hard-plastic tip was used to stimulate the midpoint of the connection between the eyes and ears of the mouse. During the experiment, the mechanical stimulation intensity increased in turn, and mouse behavior was observed at the same time. When reactions such as rubbing the mouth or scratching the head were observed, the stimulus intensity (g) was recorded. The operation was repeated five times every 30 s, and the head withdrawal threshold was recorded according to the up and down method. Subsequently, the average value was taken as the TMJ pain threshold[65].

For the open field test (OFT), a mouse was placed in a wall-enclosed square with bright light and imaged for 15 min using video recording equipment. The distance traveled, time spent in the center, and the move time were analyzed. For the elevated plus-maze test (EPM), a maze placed 60 cm above the floor was used, with a central platform, two closed arms, and two open arms. A mouse was placed on the central platform and imaged for 5 min using video recording equipment. The open arm entry, open arm time, and total number of entries were analyzed[66].

For electroencephalography, the maxillofacial regions were stimulated using a small brush 10 times, with each stimulation lasting for 5 s and an interval of 30 s. After 5 s of stimulation, the electroencephalography activity of S1BF was amplified, filtered, and recorded using an electroencephalography monitoring system (SOLAR3000N, Beijing, China).

## Statistics and reproducibility

All the data we have presented originates from distinct samples. No statistical method was used to predetermine the sample size, and all data were included in the analyses. Six biological replicates were used for statistical analysis, and representative experiments (e.g., micrographs) were from three consistent repetitions. Quantifications were performed independently and blindly by two investigators during both data collection and analysis. ImageJ software v1.48v (National Institute of Health, Bethesda, MD, USA) was used for semi-quantitative analysis. All of them were tested for normality (Shapiro-Wilk test) and equal variance assumptions (modified Levene test). If either assumption was not met, the relevant data sets were subjected to non-linear transformations. Differences among the values in more than two groups were evaluated using one-way ANOVA analysis (Tukey's post hoc test). For ordinal data, such as those obtained from experiments like the von Frey filament test, Kruskal–Wallis (KW) test followed by Dunn's test was used. All data are presented as the means ± standard deviation. $p < 0.05$ was considered statistically significant. Analyses were performed using GraphPad Prism 8.0 (GraphPad Inc., La Jolla, CA, USA).

## Reporting summary

Further information on research design is available in the Nature Portfolio Reporting Summary linked to this article.

## Data availability

All data needed to evaluate the conclusions in the paper are present in the paper and/or the Supplementary Materials. Source Data are provided with this paper. Source data is available for Figs. 2c, e–g, 3b, d–h, j, l, 4c–f, 5d–l, 6b, 7c–i, 8d–k and Supplementary Figs. 1, 2, 3b–e, 4, 5, 7.

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

## Acknowledgements

This work was sponsored by funding from National Nature Science Foundation of China (82471000 to K.J., 82325012 to L.N.) and National Key Research and Development Program (2023YFC2509100 to K.J.). We thank State Key Laboratory of Oral and Maxillofacial Reconstruction and Regeneration for technical support.

## Author contributions

K.J., L.N., and W.N. designed the study and provided funding and supervision; W.P.Q., Z.Y.M., and G.B. performed most experiments and wrote the manuscript; W.Q. and L.L. performed material preparation; D.X.H. performed Molecular dynamics simulations; Y.Z.W. prepared illustrations; J.F.Y. and X.X.H. provided technical assistance and supervision. All authors contributed to the interpretation of experiments.

## Competing interests

The authors declare no competing interests.
