## [Transparent Peer Review file · Nature Communications]

Neurovascularization inhibiting dual responsive hydrogel for alleviating the progression of osteoarthritis

Corresponding Author: Professor Kai Jiao

A version of this paper was originally rejected for publication by Nature Communications, however that decision was reconsidered after appeal by the authors.

Version 1:

Reviewer comments:

Reviewer #1

(Remarks to the Author)

Decision: Reject

In this paper, a pH and ROS responsive polycationic hydrogel (OSPPB) was formed by cross-linking aldehyde-phenylboronic acid-modified sodium alginate/polyethyleneimine-grafted protocatechuic acid (OSAP/PPCA) and bevacizumab sustained-release nanoparticles (BGN@Be). The OSPPB hydrogel functions by locally scavenging extracellular RNA and releasing bevacizumab to block neurovascularization at the osteochondral interface, thereby mitigating OA pain and disease progression. However, the results are not enough to prove the design of this study. Some typos are in red color, it seemed in an unfinished state.

1. Low pH or high ROS stimulation induced rapid hydrogel collapse and transformation into a liquid-like state in Figure 2a. However, it is difficult to quantitatively evaluate the degree of responsiveness from photographs alone.
2. The presence of elemental Si, Ca, and P in the gel matrices further confirmed BGN@Be incorporation into the OSPPB hydrogel in Figure 1e. However, OSPPB hydrogels should be used as negative controls to prove the absence of BGN@Be.

Reviewer #2

(Remarks to the Author)

In this study, an injectable ROS and pH-responsive hydrogel was synthesized to block neurovascularization by local exRNA scavenging and the release of bevacizumab, which could reduce pain and promote OA treatment. This manuscript has a clear viewpoint and sufficient data, so it can be published on Nature Communication after appropriate revisions.

There are some points that could be improved are listed below:

1. The bioactive glass nanoparticles BGN was selected as the drug carrier, but the reason for selecting BGN was not explained. It is suggested to add relevant content to state the advantages of BGN.
2. Fig. 2b showed that the OSPPB hydrogel was completely degraded in about 6 days, but the treatment period chosen in vivo was 3 weeks. Whether this will affect the treatment effect, it is suggested to give a brief explanation.
3. In Fig. 5b and Fig. S9, under the same animal model conditions, the cartilage thickness of 'UAC+OSPPB' group was much smaller than that in the other groups. Is there any difference in the sites of sampling?
4. In section 5.4.2, 'UAC+Veh' was proposed in the first paragraph, but the meaning of 'Vehicle' was appeared in the second paragraph. The definition of the abbreviation 'Veh' should be presented when it first appeared.
5. There are some other revision suggestions for this manuscript. First, compared with the hemolysis test in Fig. 2d, some data such as DPPH experiment in the Supplementary Information are more meaningful to prove the performance of hydrogels, so it is recommended to adjust. Meanwhile, Fig. 1 and Fig. 2 are suggested to be beautified. And on page 11, 'The withdrawal threshold increased significantly in UAC+Veh group' was written, but in Figure 6i, the withdrawal threshold of UAC+Veh was the smallest. There may be an error which should be modified.

Reviewer #3

(Remarks to the Author)

This paper focused on one of the primary complaints in osteoarthritis—pain. Authors constructed a novel OSPPB hydrogel for the treatment of osteoarthritis and associated pain. OSPPB hydrogel responsively releases its functional components in the low pH and high ROS microenvironment characteristic of osteoarthritis. It functions by scavenging exRNA and blocking VEGF. The study is innovative, intriguing, methodologically-sound and well-conducted. This research elucidates new disease mechanisms, and indicates new therapeutic strategies, which hold significant scientific merit and practical value. However, there are several issues that need to be addressed:

Specific comments:

1. Please carefully check the molecular chemical formulas in the Fig. 1a, as there are certain errors present, such as the molecular formula for the boronic ester bond.
2. The image quality of the self-healing experiment in Fig. 1b is not good, please replace it with a more representative image.
3. The description of Fig. 6i does not match the results shown in the image. The text states that “The withdrawal threshold increased significantly in the UAC+Veh group” but the figure shows a decrease. Please check carefully.
4. It is known that the TRPV1 channel is present on immune cells such as macrophages, which are related to pain. In this article, the degradation metabolism of the hydrogel after intra-articular injection has not been discussed, but this degradation seems closely associated with macrophages. Please expand the discussion in this article about the degradation metabolism of the hydrogel, particularly regarding the potential reactions of macrophages, and cite the following reference:
Lv Z, Xu X, Sun Z, Yang YX, Guo H, Li J, Sun K, Wu R, Xu J, Jiang Q, Ikegawa S, Shi D. TRPV1 alleviates osteoarthritis by inhibiting M1 macrophage polarization via Ca²⁺/CaMKII/Nrf2 signaling pathway. *Cell Death Dis.* 2021 May 18;12(6):504. doi: 10.1038/s41419-021-03792-8. PMID: 34006826; PMCID: PMC8131608.
5. The text uses at least four different terms to describe the same chemical bond. Please ensure consistent terminology throughout the document for “Schiff base bond” and “boronic ester bond”.
6. There are a number of errors in this article, including misuse of dashes, spelling mistake, misuse of italics, etc. For instance, in Line 4, Paragraph 1 in Part2.3, “The -LIVE/DEAD staining” is wrong

Version 2:

Reviewer comments:

Reviewer #2

(Remarks to the Author)

This manuscript synthesized an injectable ROS and pH-responsive hydrogel to reduce pain and treat OA by block neurovascularization. The validation for the design is relatively sufficient, and the suggestions presented are also modified. Therefore, this manuscript could be published on Nature Communication. Please further modify the figures, for example, in Figure 1c, the group name should not cover the curves.

Reviewer #3

(Remarks to the Author)

The authors answered all the concerns. It can be acceptable.

Responses to comments from Reviewer #1

In this paper, a pH and ROS responsive polycationic hydrogel (OSPPB) was formed by cross-linking aldehyde-phenylboronic acid-modified sodium alginate/polyethyleneimine-grafted protocatechuic acid (OSAP/PPCA) and bevacizumab sustained-release nanoparticles (BGN@Be). The OSPPB hydrogel functions by locally scavenging extracellular RNA and releasing bevacizumab to block neurovascularization at the osteochondral interface, thereby mitigating OA pain and disease progression. However, the results are not enough to prove the design of this study.

1. Low pH or high ROS stimulation induced rapid hydrogel collapse and transformation into a liquid-like state in Figure 2a. However, it is difficult to quantitatively evaluate the degree of responsiveness from photographs alone.

Our Response: Thanks for your suggestion. To address this issue, in the revised version we have included rheological measurements and Scanning Electron Microscopy (SEM) images and quantification (Fig. 2a-d).

The rheological results revealed that there were marked drop in the G' of the OSPPB hydrogel after low pH (76.76 Pa at 0.1 Hz) or high ROS stimulation (74.53 Pa at 0.1 Hz). And this drop was much more pronounced with both stimuli, with G' reducing to 17.133 Pa at 0.1 Hz. And the SEM results revealed that, while there were still visible pores in the groups subjected to either ROS (31.50% porosity) or pH (37.04% porosity) stimulation alone, the pores in the ROS + pH group (13.08% porosity) had collapsed, indicating network disintegration.

Therefore, the rheological and SEM results combining with the original images indicate that the OSPPB hydrogel exhibits responsiveness to high ROS and low pH conditions. (Fig.2a-d; Page 7, Paragraph 2, Line 4-12; Page 20, Paragraph 3, Line 4-8)

2. The presence of elemental Si, Ca, and P in the gel matrices further confirmed BGN@Be incorporation into the OSPPB hydrogel in Figure 1e. However, OSPPB hydrogels should be used as negative controls to prove the absence of BGN@Be.

Our Response: Thanks for your suggestion. To provide negative controls, we have added SEM-EDS of OSPP hydrogel and the areas of OSPPB hydrogel where BGN@Be is not distributed. The absence of Si, Ca, and P in aforementioned areas, besides their presence in the areas of OSPPB hydrogel where BGN@Be is incorporated, confirmed the incorporation of BGN@Be into the OSPPB hydrogel. We have added the data in the revised version. (Fig. 1e and Supplementary Fig. 4; Page 6, Paragraph 1, Line 9, Line 11-13)

3. The results are not enough to prove the design of this study.

Our Response: Thanks for your comment and suggestion. This study builds on our previous mechanistic research and extends into the exploration of biomaterials. Our previous data have shown that exRNAs have the ability to recruit polycationic neurovascular factors, which further amplified abnormal neurovascularization in the osteoarthritic condylar joint and resulted in unbearable pain¹. Besides, given that the osteochondral microenvironment in OA exhibits elevated levels of ROS and a decreased pH², we have designed this unique material to achieve precise intervention in the neurovascularization in OA.

Our article follows a logical sequence from material synthesis, through *in vitro* experiments, to *in vivo* testing, systematically validating the efficacy of the hydrogel. Firstly, we fabricated OSPPB hydrogel, following synthesization and characterization of three functional components, OSAP, PPCA and BGN@Be. To validate its gelation, injectability, self-healing, ROS-pH responsiveness, antioxidant properties, positive charge, and biocompatibility, we employed various methods and presented the results in Figures 1 and 2. Secondly, we demonstrated *in vitro* that the hydrogel inhibits the function of EPC and TG cells, confirming that the primary active components are PPCA and BGN@Be. Thirdly, we compared the OSPPB hydrogel with the commonly-used celecoxib in *in vivo* experiments, and confirmed its effectiveness in inhibiting OA disease progression and alleviating OA pain. This research further supports the mechanism that exRNA promotes neurovascularization and leads to OA disease progression, and provides a new therapeutic approach and offers insights into the development of inorganic-organic composite materials.

We have added the above discussion in the revised version (Page 13, Paragraph 2, Line 1-15; Page 14, Paragraph 1, Line 1-6).

References:

- 1 Qin, W. P. *et al.* Effect of extracellular ribonucleic acids on neurovascularization in osteoarthritis. *Adv. Sci.* **10**, e2301763 (2023).
- 2 Li, J., Zhang, H., Han, Y., Hu, Y., Geng, Z. & Su, J. Targeted and responsive biomaterials in osteoarthritis. *Theranostics.* **13**, 931-954 (2023).

Responses to comments from Reviewer #2

In this study, an injectable ROS and pH-responsive hydrogel was synthesized to block neurovascularization by local exRNA scavenging and the release of bevacizumab, which could reduce pain and promote OA treatment. This manuscript has a clear viewpoint and sufficient data, so it can be published on Nature Communication after appropriate revisions.

There are some points that could be improved are listed below:

1. The bioactive glass nanoparticles BGN was selected as the drug carrier, but the reason for selecting BGN was not explained. It is suggested to add relevant content to state the advantages of BGN.

Our Response: Thanks for your suggestion. In the revised version, we have added the following description to state the advantages of BGN (Page 16, Paragraph 1, Line 4-6, Line 17-18, Line 9-10).

In the present study, we chose BGN due to their controllable nanostructure, excellent biodegradability and straightforward synthesis, making them highly suitable for biomedical applications such as drug delivery and tissue engineering. Specifically, we effectively utilized the controlled drug-release properties of BGN to achieve sustained release of bevacizumab, offering an innovative approach for OA drug delivery. Furthermore, the incorporation of BGN into the polymer hydrogel provides valuable insights into the development of inorganic-organic composite materials.

2. Fig. 2b showed that the OSPPB hydrogel was completely degraded in about 6 days, but the treatment period chosen in vivo was 3 weeks. Whether this will affect the treatment effect, it is suggested to give a brief explanation.

Our Response: We greatly appreciate your insightful comments and suggestions. In our *in vivo* study, we administered a single injection and found that, three weeks later, the treatment was effective in scavenging exRNA and inhibiting neurovascularization, thereby suppressing OA progression and OA pain. During the *in vitro* studies, we observed that OSPPB hydrogel degraded within 6 days, and BGN@Be sustainedly release bevacizumab for over 30 days. Based on these data, we hypothesize that the early scavenging of exRNA and the sustained release of bevacizumab from BGN are sufficient to achieve the *in vivo* therapeutic effects. Given the differences between *in vivo* and *in vitro* environments, further *in vivo* studies are needed to monitor the real-time degradation of the hydrogel in an OA environment characterized by high ROS and low pH to further validate the present conclusions.

We have added the above discussion in the revised version (Page 17, Paragraph 1, Line 1-3)

3. In Fig. 5b and Fig. S9, under the same animal model conditions, the cartilage thickness of 'UAC+OSPPB' group was much smaller than that in the other groups. Is there any difference in the sites of sampling?

Our Response: Thanks for your suggestion. As you pointed out, due to the

heterogeneity of the specimens, the images in Fig. 5b and Fig. S9 were not representative. We have replaced them by the more representative figures to ensure accurate comparison in the revised version. (Fig. 5b and Fig. S9)

4. In section 5.4.2, 'UAC+Veh' was proposed in the first paragraph, but the meaning of 'Vehicle' was appeared in the second paragraph. The definition of the abbreviation 'Veh' should be presented when it first appeared.

Our Response: Thanks for your suggestion. We have defined "Veh" in the first paragraph of Section 5.4.2, clarifying that we used PBS as the vehicle. (Page 23, Paragraph 4, Line 2)

5. There are some other revision suggestions for this manuscript. First, compared with the hemolysis test in Fig. 2d, some data such as DPPH experiment in the Supplementary Information are more meaningful to prove the performance of hydrogels, so it is recommended to adjust.

Our Response: Thanks for your suggestion. We have rearranged the positions of the DPPH experiment and the hemolysis test. The hemolysis test is now placed in the Supplementary Information, while the DPPH experiment is included in Figure 2g. (Fig.2g and Supplementary information)

6. Meanwhile, Fig. 1 and Fig. 2 are suggested to be beautified.

Our Response: Thanks for your suggestion. We have beautified Figures 1 and 2 by adding a background color to the schematic diagrams, standardizing the font and font size, aligning the images, and replacing low-quality images with higher-quality ones. (Fig. 1 and Fig. 2)

7. And on page 11, ‘The withdrawal threshold increased significantly in UAC+Veh group’ was written, but in Figure 6i, the withdrawal threshold of UAC+Veh was the smallest. There may be an error which should be modified.

Our Response: Thanks for your suggestion. We have verified and corrected the inaccurate description in the text. The correction is: "The withdrawal threshold decreased significantly in the UAC+Veh group." (Page 11, Paragraph 2, Line 16-17)

Responses to comments from Reviewer #3

This paper focused on one of the primary complaints in osteoarthritis—pain. Authors constructed a novel OSPPB hydrogel for the treatment of osteoarthritis and associated pain. OSPPB hydrogel responsively releases its functional components in the low pH and high ROS microenvironment characteristic of osteoarthritis. It functions by scavenging exRNA and blocking VEGF. The study is innovative, intriguing, methodologically-sound and well-conducted. This research elucidates new disease mechanisms, and indicates new therapeutic strategies, which hold significant scientific merit and practical value. However, there are several issues that need to be addressed:

Specific comments:

1. Please carefully check the molecular chemical formulas in the Fig. 1a, as there are certain errors present, such as the molecular formula for the boronic ester bond.

Our Response: Thanks for your suggestion. We have reviewed the molecular chemical formulas in Fig. 1a and have corrected the molecular formula for the boronic ester bond. (Fig. 1a)

2. The image quality of the self-healing experiment in Fig. 1b is not good, please replace it with a more representative image.

Our Response: Thanks for your suggestion. We have now replaced original images with more representative and higher-quality ones. (Fig. 1b)

3. The description of Fig. 6i does not match the results shown in the image. The text states that “The withdrawal threshold increased significantly in the UAC+Veh

group” but the figure shows a decrease. Please check carefully.

Our Response: Thanks for your suggestion. We have verified and corrected the inaccurate description in the text. The correction is: "The withdrawal threshold decreased significantly in the UAC+Veh group." (Page 11, Paragraph 2, Line 16-17)

4. It is known that the TRPV1 channel is present on immune cells such as macrophages, which are related to pain. In this article, the degradation metabolism of the hydrogel after intra-articular injection has not been discussed, but this degradation seems closely associated with macrophages. Please expand the discussion in this article about the degradation metabolism of the hydrogel, particularly regarding the potential reactions of macrophages, and cite the following reference:

Lv Z, Xu X, Sun Z, Yang YX, Guo H, Li J, Sun K, Wu R, Xu J, Jiang Q, Ikegawa S, Shi D. TRPV1 alleviates osteoarthritis by inhibiting M1 macrophage polarization via Ca²⁺/CaMKII/Nrf2 signaling pathway. Cell Death Dis. 2021 May 18;12(6):504. doi: 10.1038/s41419-021-03792-8. PMID: 34006826; PMCID: PMC8131608.

Our Response: Thanks for your suggestion. Due to the current inability to perform real-time monitoring of ROS, pH, and hydrogel degradation, we further did not discuss the degradation metabolism of the hydrogel. We apologize for this oversight, which may have misled you and caused misunderstanding. Our study primarily focused on sensory neurons and vascular endothelial cells but lacked consideration of other cell types. However, as macrophages are part of the innate immune system, they likely participate in the degradation of the hydrogel and may influence downstream OA pain through polarization. And we have now cited the recommended literature to address this aspect more comprehensively. (Page 17, Paragraph 1, Line 4-7)

5. The text uses at least four different terms to describe the same chemical bond. Please ensure consistent terminology throughout the document for “Schiff base bond” and “boronic ester bond”.

Our Response: Thanks for your suggestion. We have standardized the expressions for "Schiff base bond" and "boronic ester bond" throughout the manuscript. (Page 1, Paragraph 1, Line 11-12; Page 3, Paragraph 2, Line 8-9; Page 4, Paragraph 3, Line 4-6; Page 5, Paragraph 2, Line 3; Page 6, Paragraph 2, Line 1; Page 7, Paragraph 2, Line 17; Page 13, Paragraph 2, Line 4, Line 6; Page 15, Paragraph 2, Line 7-8; Page 16, Paragraph 2, Line 11; Page 17, Paragraph 2, Line 6)

6. There are a number of errors in this article, including misuse of dashes, spelling mistake, misuse of italics, etc. For instance, in Line 4, Paragraph 1 in Part2.3, “The -LIVE/DEAD staining” is wrong

Our Response: Thanks for your suggestion. We have reviewed the entire manuscript and corrected various minor errors, including the "LIVE/DEAD staining." (Page 8, Paragraph 3, Line 4)

Responses to comments from Reviewer #2

This manuscript synthesized an injectable ROS and pH-responsive hydrogel to reduce pain and treat OA by block neurovascularization. The validation for the design is relatively sufficient, and the suggestions presented are also modified. Therefore, this manuscript could be published on Nature Communication. Please further modify the figures, for example, in Figure 1c, the group name should not cover the curves.

Our Response: Thanks for your suggestion. We have checked all figures and modified the Figure 1c, Figure S1 and Figure S2.

Figure 1c,

Figure S1,

Figure S2,